# A Bayesian framework for inferring regional and global change from stratigraphic proxy records (StratMC v1.0)

Stacey Edmonsond[1] and Blake Dyer[1]

[1]School of Earth and Ocean Sciences, University of Victoria, Victoria, BC, Canada

**Correspondence:** Stacey Edmonsond (sedmonsond@uvic.ca)

**Abstract.** The chemistry of ancient sedimentary rocks encodes information about past climate, element cycling, and biological innovations. Records of large-scale Earth system change are constructed by piecing together geochemical proxy data from many different stratigraphic sections, each of which may be incomplete, time-uncertain, biased by local processes, and diagenetically altered. Accurately reconstructing past Earth system change thus requires correctly correlating sections from different locations, distinguishing between global and local changes in proxy values, and converting stratigraphic height to absolute time. Incomplete consideration of the uncertainties associated with each of these challenging tasks can lead to biased and inaccurate estimates of the magnitude, duration, and rate of past Earth system change. Here, we address this shortcoming by developing a Bayesian statistical framework for inferring the common proxy signal recorded by multiple stratigraphic sections. Using the principle of stratigraphic superposition and both absolute and relative age constraints, the model simultaneously correlates all stratigraphic sections, builds an age model for each section, and untangles global and local signals for one or more proxies. Synthetic experiments confirm that the model can correctly recover proxy signals from incomplete, noisy, and biased stratigraphic observations. Future applications of the model to the geologic record will enable geoscientists to more accurately pose and test hypotheses for the drivers of past proxy perturbations, generating new insights into Earth's history. The model is available as an open-source Python package (*StratMC*), which provides a flexible and user-friendly framework for studying different times and proxies recorded in sediments.

## 1 Introduction

Sedimentary rocks host fragments of information about the long-term co-evolution of Earth's surface environments, biosphere, and lithosphere. Much of this information is encoded by geochemical proxies that are used to make inferences about past changes in one or more Earth systems. For example, the sulfur isotopic composition of sulfate and sulfide minerals tracks Earth's redox evolution (Farquhar et al., 2000), while measurements of the carbon isotopic composition of carbonate sediments have been used to identify past perturbations to Earth's surface carbon cycle (Kump and Arthur, 1999). Since most deep-sea sediments older than ∼200 Ma have been subducted at continental margins, reconstructions of Earth system change prior to the mid-Mesozoic rely primarily on observations from sediments deposited in marginal shallow-water environments that escape subduction.

Records of average large-scale change are constructed by placing proxy data from many different locations in the same chronostratigraphic reference frame. In practice, this placement typically is achieved by considering both relative and absolute age constraints. Relative age models are constructed using some combination of bio-, litho-, and chemostratigraphy, where fossil occurrences, marker beds, and geochemical trends are correlated among stratigraphic sections. Where available, correlation is guided by geochronological age constraints (e.g., radiometrically dated ash beds or detrital minerals). Once all observations have been placed in the same relative reference frame, geochronological ages are used to construct an absolute age model, where each observation is mapped to time. The timing, duration, rate, and environmental context of large-scale proxy change then can be evaluated using the combined data from all locations.

It often is exceedingly challenging to construct accurate age models – and, consequently, accurate estimates of proxy change over time – owing to three fundamental features of the shallow-water stratigraphic record. First, the stratigraphic record is incomplete: sedimentary sequences are punctuated by hiatuses (surfaces that represent non-deposition or erosion), and any single location may preserve only a few disjoint fragments of geologic time (Sadler, 1981). Consequently, the relationship between time and stratigraphic height often is complex and irregular. Second, materials amenable to geochronological dating are rare in ancient sedimentary strata, limiting the resolution of absolute age models. Third, many geochemical proxies can be influenced by local and post-depositional processes. For example, the $\delta^{13}$C of shallow-water carbonate sediments, which frequently is used for correlation of unfossiliferous Precambrian sediments (Knoll et al., 1986; Halverson et al., 2005; Xiao et al., 2016; Bowyer et al., 2022; Halverson et al., 2022; Topper et al., 2022), commonly is at least partly decoupled from the $\delta^{13}$C of contemporaneous global-mean seawater dissolved inorganic carbon (DIC) due to both primary processes (e.g., local biological activity; Patterson and Walter, 1994; Swart, 2008; Geyman and Maloof, 2019; Trower et al., 2024) and diagenesis (Allan and Matthews, 1982; Higgins et al., 2018). In addition to complicating correlation among sections, these local processes may obscure the true nature of proxy change over time by driving stratigraphic changes in proxy values that are unrelated to large-scale biogeochemical cycling.

Many stratigraphers have recognized the challenges associated with constructing both relative and absolute age models, and a host of quantitative tools designed to treat unknowns in a more explicit and reproducible way has emerged in response (e.g., Hagen, 2024). These tools include both classical approaches, which rely only on observed data (e.g., geochronological ages with uncertainties), and Bayesian approaches, which also explicitly consider *a priori* knowledge about the system of interest (e.g., superposition constraints). In the realm of absolute age model construction, the widespread adoption of Bayesian methods has led to more conservative estimates of uncertainty in the ages of undated stratigraphic horizons (e.g., Johnstone et al., 2019; Trayler et al., 2020; Halverson et al., 2022; Zhang et al., 2023). Meanwhile, dynamic time warping – a deterministic algorithm for finding the optimal least-squares alignment between two sequences – has been used for stratigraphic correlation of carbonate $\delta^{13}$C (Hay et al., 2019; Ajayi et al., 2020; Hagen and Creveling, 2024), paleomagnetic (Hagen et al., 2020; Peti et al., 2020; Reilly et al., 2023), ice core (Hagen and Harper, 2023), and borehole well data (Baville et al., 2022; Sylvester, 2023). Various correlation algorithms also have been developed by the deep-sea sediment core community for application to more continuous Cenozoic $\delta^{13}$C and $\delta^{18}$O records (e.g., Lisiecki and Lisiecki, 2002; Lin et al., 2014; Ahn et al., 2017; Lee et al., 2023). Notably, the Bayesian approach of Lee et al. (2023) improves on previous algorithms by enforcing radiometric age constraints

during the alignment step and iteratively aligning all records to a composite proxy 'stack' rather than to a single target record. However, that approach relies on a prior model for sedimentation rate that is only appropriate for deep-sea sediment cores. Most recently, Bloem and Curtis (2024) developed a Bayesian approach to intrabasinal chemostratigraphic correlation underpinned by computational simulations of sediment accumulation. This model quantifies uncertainty in the alignment among sections, but can only be used to correlate sections within a single basin and does not consider local influences on proxy values.

Together, all of the methods described above constitute an important step toward an objective and reproducible approach to correlation. However, none are well-equipped to handle observations from ancient shallow-water environments where age constraints are sparse and proxy data may be locally biased, diagenetically altered, and incomplete. Furthermore, no existing chemostratigraphy algorithm is specifically optimized for reconstructing past changes in global biogeochemical cycling. Instead, correlation is the main objective while proxy change over time is reconstructed subsequently, typically by stacking the aligned proxy observations (e.g., Lisiecki and Raymo, 2005; Hagen and Creveling, 2024). In the context of reconstructing past Earth system change, this focus on correlation neglects that the observed proxy values may have been influenced by many processes other than global biogeochemical cycling.

Here, we address this gap by developing a new Bayesian framework for inferring the common proxy signal recorded by multiple stratigraphic sections using only age constraints and the principle of stratigraphic superposition. This model 1) explicitly attempts to deconvolve global and local signals, 2) simultaneously correlates all stratigraphic sections using both relative and absolute age constraints, 3) constructs a distribution of age models for each section during the correlation step, and 4) can simultaneously infer global changes in multiple proxies. Importantly, this modeling approach does not replace the need for geologists to carefully consider the geology and broader context of the systems they are reconstructing. It does, however, provide a quantitative framework for testing hypotheses and instilling geologic wisdom into the proxy signal reconstruction process. We demonstrate the method using synthetic carbonate $\delta^{13}$C data, and then leverage simple experiments to explore the broader implications for reconstructing the history of past Earth systems using stratigraphic proxy records.

## 2 Bayesian inference model

### 2.1 Overview

In the following subsections, we develop a Bayesian statistical framework to find the proxy history that can best describe a given set of stratigraphic observations (Fig. 1). Details of the model structure are given in Sect. 2.3 through 2.6. In Sect. 2.7 and 2.8, we provide additional practical guidance for using the model; more extensive instructions are provided in the supplementary *User Manual* and package documentation (https://stratmc.readthedocs.io/).

Here, we briefly summarize key aspects of the model structure. The model requires two inputs: proxy observations from multiple stratigraphic sections and age constraints (at least a minimum and maximum age for each input section, reported with uncertainties). The model-derived age of each proxy observation obeys age constraints and respects superposition with all other observations from the same section. Given these constraints, the model extracts the *shared component* of the proxy signal recorded by all stratigraphic sections. The form and timing of this shared component is learned from the data. Throughout the

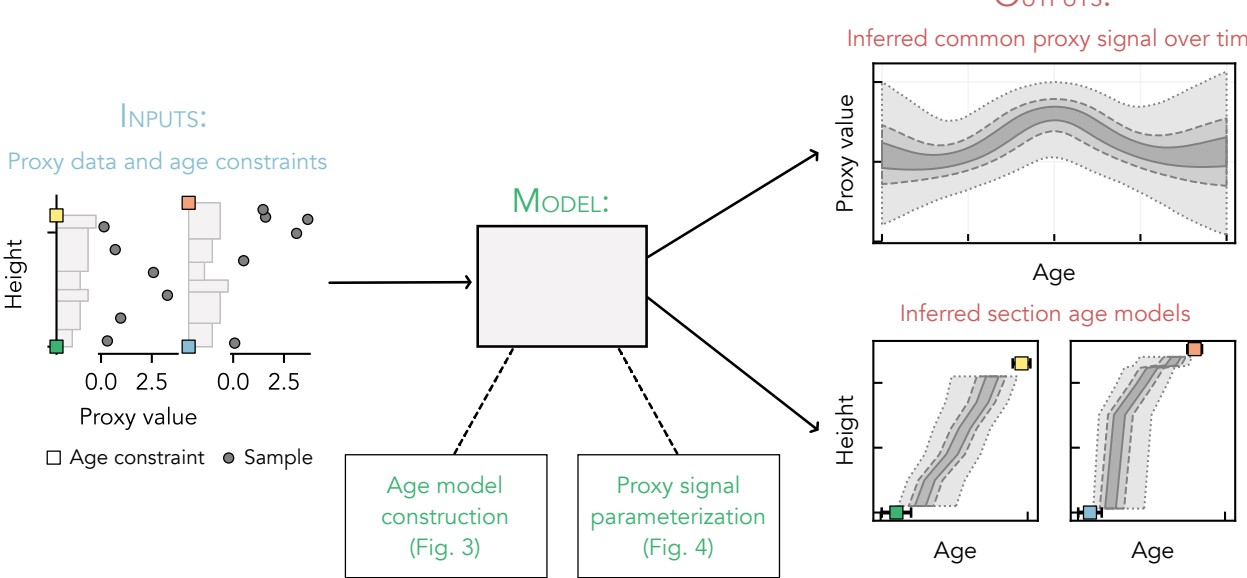

**Figure 1.** Overview of the inference model workflow. Stratigraphic proxy observations and age constraints are input to the model. Using these input data, the model simultaneously infers the common proxy signal and an age model for each section (the outputs). Key aspects of the internal model structure are illustrated in Figs. 3 and 4.

text, we often refer to this shared component as the 'global signal'. However, it may describe common change at any scale (e.g., intrabasinal, regional, or global), depending on the geographic distribution of the observations. The model also infers the degree to which individual sections are influenced by localized processes (e.g., diagenesis).

## 2.2   Bayes' Theorem

Bayesian models seek to infer the value of unknown parameters of interest ($\theta$) – for example, the age of a sample or the value of a proxy signal over time – by conditioning *a priori* knowledge about these parameters on observed data ($\mathcal{D}$). The posterior probability of the parameters conditioned on the data, $P(\theta|\mathcal{D})$, is described by Bayes' theorem:

$$P(\theta|\mathcal{D}) = \frac{P(\mathcal{D}|\theta)P(\theta)}{P(\mathcal{D})} \tag{1}$$

The model *prior*, P($\theta$), is a probabilistic representation of our existing knowledge about the parameters. For example, the prior age for a geologic sample may be constrained by overlying and underlying geochronological age constraints via the principle of stratigraphic superposition. The *likelihood*, $P(\mathcal{D}|\theta)$, is the probability of observing the data, $\mathcal{D}$, given this prior knowledge. Finally, the *evidence* or *marginal likelihood*, $P(\mathcal{D})$, is the average probability of the data with respect to the prior. The $P(\mathcal{D})$ term is constant for a given model, and is generally ignored because it is intractable to compute. Instead, we use Markov Chain

Monte Carlo (MCMC) methods (Sect. 2.7) to draw random samples from the posterior, which is proportional to the product of the likelihood and the prior.

## 2.3 Model inputs

The model requires two inputs (Fig. 1): proxy observations from multiple stratigraphic sections and age constraints (at least a minimum and maximum age for each input section). Uncertainties associated with the input data are propagated through all subsequent calculations.

## 2.4 Modeling age constraints

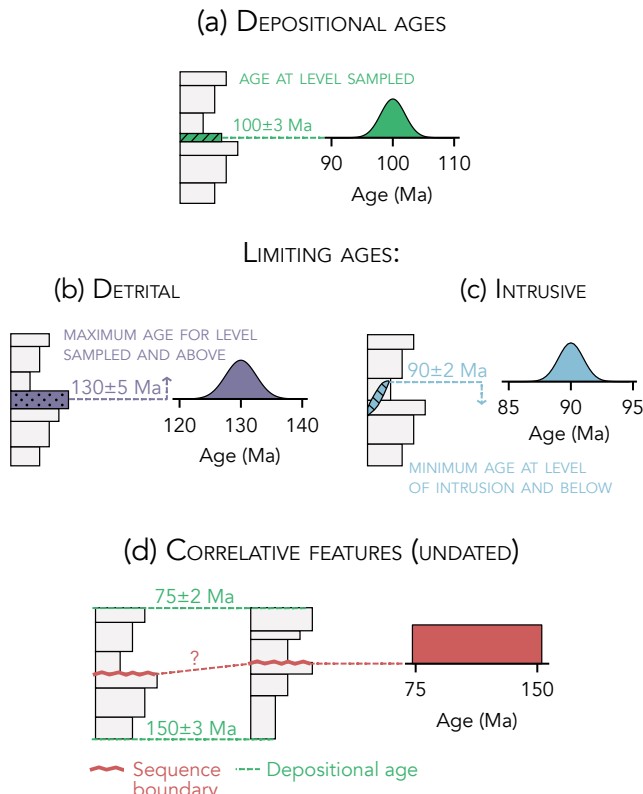

**Figure 2.** Types of age constraints that can be incorporated in the inference model. Depositional ages *(a)* directly constrain the age of strata at the dated horizon, while limiting ages *(b-c)* indirectly constrain the age of overlying (detrital age) or underlying (intrusive age) strata. Correlative features *(d)*, such as sequence boundaries or marker beds, constrain the alignment between sections. Correlative features may be undated or linked to a depositional/limiting geochronological age. In this example, an undated sequence boundary is modeled as a uniform distribution bounded by the mean reported ages of over- and underlying depositional age constraints.

Two types of absolute age constraint can be incorporated in the model: *depositional* ages (e.g., radiometrically dated ash beds), which directly date the deposition of a particular stratigraphic horizon (Fig. 2a), and *limiting* ages, which indirectly constrain the age of deposition. Examples of limiting age constraints include ages for detrital minerals (i.e., minerals derived from the erosion of pre-existing rocks), which provide a maximum age constraint at the level sampled and above (Fig. 2b), and ages for intrusive dykes and sills, which provide a minimum age constraint at the level of intrusion and below (Fig. 2c).

Age constraints are modeled using probability distributions that most accurately reflect their reported value and any associated uncertainties. By default, age constraints are modeled as normal distributions with mean and standard deviation equal to the reported age and its uncertainty (Fig. 2a-b). However, custom prior distributions can be specified to model non-Gaussian uncertainties. For example, a biozone boundary that has not been dated directly, but that has a known minimum and maximum age, could be modeled as a uniform distribution.

*Correlative features* are distinct stratigraphic horizons – such as marker beds, biozone boundaries, or sequence boundaries – that are present in multiple sections. Correlative features are modeled such that overlying samples must be younger than the feature everywhere, and underlying samples must be older than the feature everywhere. In other words, superposition with respect to the feature is universally enforced, even though the feature itself may span a slightly different interval of time in different locations (e.g., a time-transgressive sequence boundary). Both dated and undated correlative features add information by constraining the alignment between sections. For example, a sequence boundary that has been identified in two sections from the same basin, where both sections only have minimum and maximum depositional age constraints of $75 \pm 2$ and $150 \pm 3$ Ma, can be modeled as a uniform distribution bounded by $75 - 2$ and $150 + 3$ Ma (Fig. 2c). On the other hand, a correlative feature that has been dated directly is modeled in the same way as any other geochronological age (Fig. 2a-b), with the additional condition that it must have the same age in all sections.

## 2.5 Modeling sample ages

The inference model assumes there is a *common component* to the signal (proxy value over time) recorded by all stratigraphic sections. The proxy values recorded by any given stratigraphic section may reflect the convolution of this common, or 'global', signal and various non-global signals (e.g., local biogeochemical cycling and diagenesis) that are not shared by all sections. Using this assumption and all available age constraints (Sect. 2.4), it simultaneously infers an age model for each section and the common proxy signal.

In order to infer the common proxy signal, each section must have a prior age model. We construct these prior age models with the goal of imposing no limits on sedimentation rate between age constraints, meaning that the possible depositional histories for each section range from highly episodic to uniform (Fig. 3c). The prior likelihood of different depositional histories also should reflect our knowledge about how sediment accumulates in nature: namely, that extremely large and rapid depositional events are rare compared to more gradual sedimentation (Sadler, 1981). To achieve this, we construct prior distributions for the age of each proxy observation, or sample, by assuming only that age decreases with height in each section (stratigraphic superposition). Under the superposition assumption, the age of each sample is limited by its bounding age constraints (underlying age $T_1$ and overlying age $T_2$; Fig. 3b). For a given realization, the interval of time spanned by the samples (the gray shaded

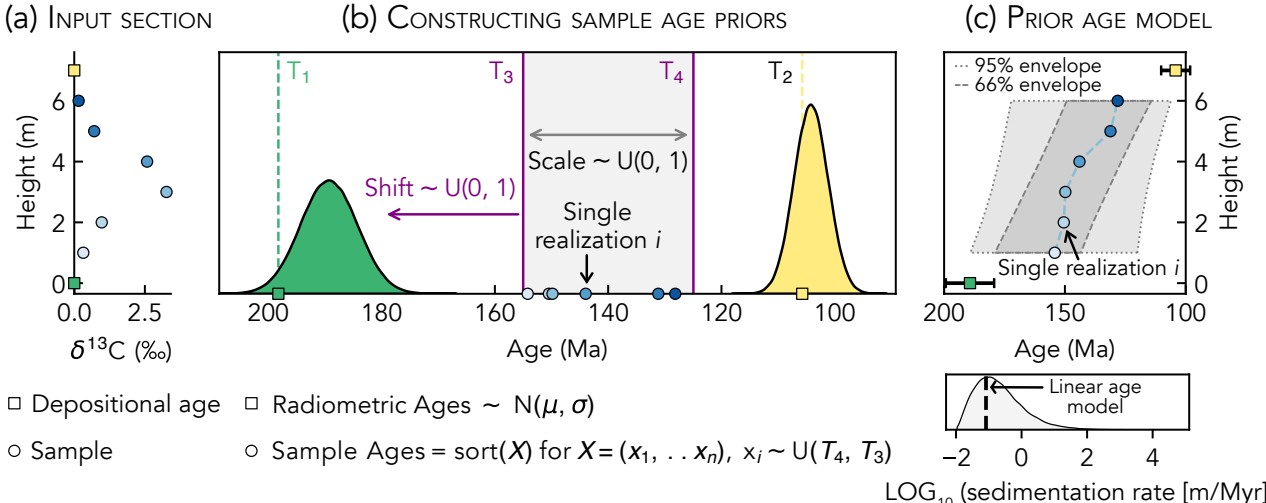

**Figure 3.** *(a)* $\delta^{13}$C observations and age constraints (model *inputs*) for a hypothetical stratigraphic section. *(b)* Procedure for constructing a prior age model for the section in *a*; see full explanation in Sect. 2.5 of the main text. *(c)* Prior section age model resulting from the procedure illustrated in *b*. The 95% envelope marks the 2.5th and 97.5th percentiles of the posterior age distribution for each sample, while the 66% envelope marks the 17th and 83rd percentiles. The example realization is the same as in *b*. The lower panel shows the probability distribution of apparent sedimentation rates (calculated between pairs of adjacent samples) resulting from this prior age model.

region in Fig. 3b) is controlled by the *scale* and *shift* parameters. The scale parameter controls the fraction of the total available time ($T_1$-$T_2$) spanned by the samples (the width of the box in Fig. 3b; $T_3$-$T_4$), while the shift parameter slides this window forward and backward in time. We place uniform prior distributions on the scale and shift parameters. Sample age priors are

constructed using sorted draws from uniform distributions bounded by $T_4$ and $T_3$, where sorting ensures that sample ages decrease upsection (Fig. 3c). The resulting prior age models encompass all possible depositional histories, but assign lower prior probabilities to solutions that are geologically unlikely. For example, extremely rapid deposition of the entire section (the far right tail of the prior sedimentation rate distribution; Fig. 3c) is less likely than more gradual deposition. The prior age models also are consistent with Sadler's (1981) empirical model of how time is distributed in stratigraphy: sedimentation

rates are approximately log-normally distributed (Fig. 3c), and each section's age model exhibits a power-law scaling between timespan and apparent sedimentation rate (i.e., the Sadler effect). The posterior age models are computed by merging these prior expectations with evidence in the data (Eq. 1).

   Limiting age constraints located in the middle of a section must be enforced in a different way to ensure that superposition is respected. For example, consider a section that is bounded by two dated ash beds ($T_1$ and $T_2$), and that also has an intermediate

detrital zircon age ($T_{1.5}$). This detrital age provides a maximum age for all overlying samples, but does not constrain the age of underlying samples (Fig. 2b). Therefore, age priors for samples below the detrital constraint would be bounded by $T_1$ and $T_2$, while age priors for samples above the detrital constraint would be bounded by $T_{1.5}$ and $T_2$. As a result, superposition

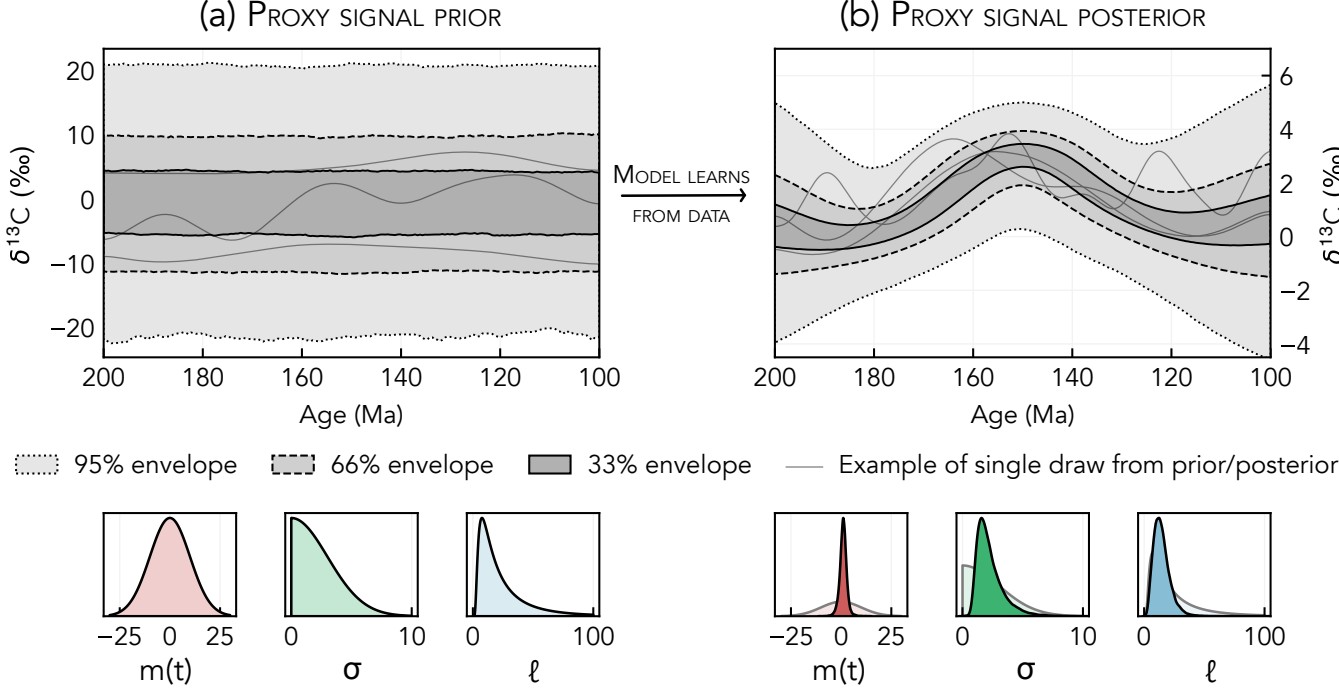

**Figure 4.** *(a)* Prior distribution for the $\delta^{13}$C signal over time before the data are considered. The 95% envelope marks the 2.5th and 97.5th percentiles of the prior $\delta^{13}$C distribution at each age, the 66% envelope marks the 17th and 83rd percentiles, and the 33% envelope marks the 33.5th and 66.5th percentiles. The prior includes a wide range of signals with different shapes and frequencies. The lower panel shows the associated priors for the Gaussian process mean function ($m(t)$), the RBF kernel variance ($\sigma$), and the RBF kernel lengthscale ($\ell$). *(b)* The posterior distribution for the $\delta^{13}$C signal over time is calculated by conditioning the prior on the data (proxy observations and age models). The lower panel shows the posterior distributions for the Gaussian process parameters; prior distributions (as in *a*) are plotted for comparison.

between samples would not be strictly enforced. To circumvent this issue, we instead construct sample age priors using only depositional age constraints (or limiting age constraints that apply to the entire section), and enforce intermediate limiting age constraints by explicitly penalizing the model likelihood when a limiting age constraint is violated. This penalty is large enough that the posterior will never include age models that violate limiting age constraints.

In some cases, different sections may have a known stratigraphic relationship (based on e.g., regional mapping of geological formations) that is not reflected by the available age constraints. To explicitly enforce known superposition relationships between such sections, the uppermost (youngest) sample from the older section is used as the maximum age constraint for the younger section ($T_1$ in Fig. 3b).

## 2.6 Modeling proxy signals

### 2.6.1 Gaussian process regression

Using the prior age of each sample, the relationship between time, $t$, and the common proxy signal, $f(t)$, is modeled as a Gaussian process. A Gaussian process (GP) defines a distribution of random functions that are described by their mean, $m(t)$, and covariance, $k(t, t')$ (Rasmussen et al., 2006) (Eq. 2).

$$f(\text{t}) \sim GP(m(t), k(t, t'))$$ (2)

We specify the GP prior such that before any data are considered, the common proxy signal can take any functional form (from e.g., uncorrelated noise to linear) (Fig. 4a). The magnitude of variance and lengthscale of covariance are inferred directly from the data. To accomplish this, we set the GP covariance function to the sum of a radial basis function (RBF) kernel and a white noise kernel with variance equal to $0.1$. We define the GP mean function as a constant, and set the prior for this constant to a normal distribution with $\mu$ and $\sigma$ chosen to encompass the full range of observed proxy values.

The prior for the RBF kernel lengthscale and variance should be tuned based on the timescale of interest and the magnitude of observed variability in the proxy data (see Sect. 4.4.1 for extended guidance). For example, in Fig. 4a, the prior for the RBF kernel variance is a half-normal (positive only) distribution with $\sigma = 3$, while the lengthscale prior is a Wald distribution with $\mu = 25$ and $\lambda = 25$. This lengthscale prior ignores high-frequency 'noise' that may be superimposed on the long-term signal, while the variance prior excludes changes in $\delta^{13}C$ that are much larger than the observed range in the data.

For data sets that include two or more proxies, each proxy signal is modeled as a separate GP with a unique prior. The GP prior for each proxy signal should be specified based on its observed variance and relevant geologic context (e.g., the proxy residence time or characteristic timescale for a process of interest).

### 2.6.2 Incorporating local variations in proxy records

We assume that each section may be influenced by 'geologic noise' from processes unrelated to the common signal (e.g., local water column processes and diagenesis), and that the proxy value recorded by each sample may be shifted relative to the common signal. To encode this assumption, the proxy value for each sample ($y_{\text{sample}}$) is modeled as a normal distribution with a mean equal to the sum of the Gaussian process evaluated at the sample age (Eq. 2) and an *offset* term ($\phi$), and standard deviation equal to the sum of measurement uncertainty ($\sigma_{\text{sample}}$) and a per-section *geologic noise* term ($\eta_{\text{section}}$) (Eq. 3). The following two paragraphs further describe the offset and noise terms, respectively.

$$y_{\text{sample}} \sim Normal(f(t) + \phi, \sigma_{\text{sample}} + \eta_{\text{section}})$$ (3)

The offset term ($\phi$) ensures that sections which covary with the common signal, but that have different absolute proxy values, are still correctly aligned. In general, we recommend using a per-section offset term, which encodes the assumption that the offset within each section is constant over time. However, alternative offset parameterizations may be appropriate in cases where 1) offset should be excluded (i.e., $\phi = 0$) because local offsets from the common signal are strictly not expected for

the proxy of interest (e.g., $^{87}$Sr/$^{86}$Sr), or 2) the geologic context supports alternative offset groupings. For example, because carbonate $\delta^{13}$C often varies among different depositional environments, we might parameterize offset such that all samples from the same environment and basin share an offset term (as in Sect. 3.2.3). The choice of offset groupings is further discussed in Sect. 4.3.4. The default offset prior is a Laplace distribution with $\mu = 0$ and $b = 2$, which assigns the highest prior probability to solutions with no offset ($\phi = 0$) and has fat tails (high kurtosis) that allow for a wide range of offset values. This prior can be modified based on the range of reasonable offset values for different proxies.

The per-section geologic noise term, $\eta_{\text{section}}$, accounts for deviations from the common signal that are not captured by the offset term. For example, diagenesis could either homogenize or increase the variance of proxy values within a given section. The default prior for $\eta_{\text{section}}$ is a half-Cauchy (positive only) distribution with $\beta = 1$, which gives the highest prior probability to solutions without local geologic noise ($\eta_{\text{section}} = 0$) but has a fat tail (high kurtosis) that allows for high noise values.

## 2.7  Sampling the posterior

### 2.7.1  Markov chain Monte Carlo sampling

The posterior distributions of all model parameters are sampled using the No-U-Turn Sampler (Hoffman and Gelman, 2014), which is a MCMC method. For each experiment in Sect. 3, the posterior is sampled by 100 independent Markov chains. Each simulation is run for $2,000$ steps, and the first $1,000$ samples (which are used to tune the sampler during the 'burn-in' period) are discarded. Rare outlier chains with extremely low posterior likelihoods (indicative of poor tuning) were discarded. All model and sampling code is implemented in the Python probabilistic programming package PyMC (Abril-Pla et al., 2023).

### 2.7.2  Assessing convergence

A MCMC sampling algorithm has *converged* when each Markov chain stabilizes to the same posterior distribution (Fig. A1b). MCMC algorithms can struggle to converge when sampling complex and multimodal posterior distributions (e.g., proxy observations with multiple possible ages) because each Markov chain gets 'stuck' in a single mode, resulting in incomplete exploration of the parameter space (Fig. A1a). Consequently, different chains sometimes do not converge on the same posterior proxy signal. While various algorithms for improving exploration of multimodal posteriors have been developed (e.g., parallel tempering; Earl and Deem, 2005), they are often computationally expensive and difficult to tune.

Here, we mitigate convergence issues for models with complex posterior distributions by running many Markov chain simulations (each of which may explore a different mode of the posterior) in parallel. When the model posterior has stabilized – meaning that incorporating additional chains does not affect the results – we consider the posterior to be sufficiently well-explored. In Appendix A, we develop specific tests for evaluating whether the posterior is stable. While stability always should be assessed before interpreting the results, running 20 chains in parallel is adequate for most models.

## 2.8 Working with large data sets

The computational complexity of exact Gaussian process inference scales as $\mathcal{O}(n^3)$, where $n$ is the number of observations (Rasmussen et al., 2006). As a result, inference quickly becomes intractable for more than several hundred proxy observations. In brief, we propose two possible approaches for working with large data sets. The first approach – 'data downscaling' – is to reduce the number of proxy observations included in the inference in a way that does not significantly influence the results (e.g., removing sections that are redundant or that have poor age constraints). Alternatively, the proxy signal can be modeled using an approximate Gaussian process (e.g., Riutort-Mayol et al., 2022) instead of an exact GP. We elaborate on the GP approximation approach in Appendix B.

## 3 Case studies

We apply the inference model to computer-generated proxy data and age constraints. In each experiment, posteriors generated using synthetic stratigraphic data are compared to a known proxy signal encoded in the data. The similarity between the inferred and known proxy signal measures how accurately the model can reconstruct proxy signals from stratigraphic data with different prescribed characteristics (e.g., local biases or complex age models).

Experiments are conducted following the methodology outlined in Sect. 3.1. First, we run simple tests to verify that the inference model can successfully recover one or more known proxy signals using only stratigraphic observations (Sect. 3.2). Then, we demonstrate how the model performs when applied to different types of incomplete and locally biased data that may appear in the rock record (Sect. 3.3). More generally, these experiments highlight both the utility and challenges of using chemostratigraphic data to reconstruct past Earth system change.

### 3.1 Methodology

#### 3.1.1 Generating synthetic data for experiments

In Sect. 3.2, we conduct basic tests of the inference model using synthetic data that are generated in two steps (Fig. 5). First, we define an imaginary proxy signal over time (Fig. 5a). In this example, the proxy signal is the $\delta^{13}$C of global-mean seawater DIC from 450 to 400 Ma. Second, we translate this synthetic signal to the rock record using a range of computer-generated age models (Fig. 5b). In total, the synthetic data set includes 171 $\delta^{13}$C observations distributed among six stratigraphic sections. The $\delta^{13}$C observations are assigned measurement uncertainties ($\sigma_{\text{sample}}$ in Eq. 3) of 0.1‰. All sections are bounded by the same maximum ($450 \pm 2.4$ Ma, $2\sigma$) and minimum ($400 \pm 2.1$ Ma, $2\sigma$) depositional age constraints.

For each experiment, we modify these synthetic data in two ways. First, we add Gaussian noise to the $\delta^{13}$C observations in order to simulate random natural variability and/or specific geologic processes (e.g., local carbon cycling and diagenesis) that can partially decouple the $\delta^{13}$C of carbonate rocks from that of contemporaneous global-mean seawater DIC. Second, we modify the age constraints and lithostratigraphy for each section. The lithostratigraphy is modified either to support new age

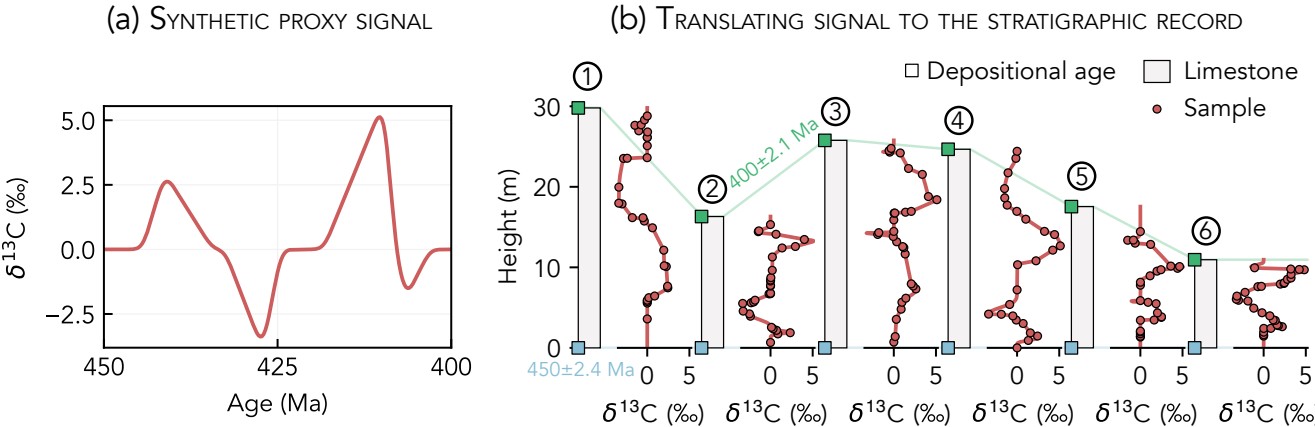

**Figure 5.** Procedure for generating the synthetic proxy data used in Sect. 3.2. A synthetic proxy signal *(a)* is translated to the stratigraphic record *(b)* using six computer-generated age-height models. While all synthetic sections share the same maximum and minimum age constraints, each section has a unique depositional history.

constraints (e.g., inserting correlative marker beds; Sect. 3.2.1), or to aid in simulating environment-dependent geochemical variability (Sect. 3.2.3).

In Sect. 3.3, we conduct a second set of experiments that aim to quantify the effect of noise on the accuracy of proxy signal reconstructions. We simulate both 'proxy noise', which emulates local processes that increase the variance of preserved proxy

values, and 'temporal noise', where non-uniform depositional histories produce irregular age-height relationships. For each experiment, we measure the effect of noise amplitude on proxy signal recovery.

Depositional histories ranging from continuous to episodic are simulated following the procedure in Fig. 6. First, we define a synthetic $\delta^{13}$C signal from 130 to 100 Ma (Fig. 6a). Then, we translate this signal to four stratigraphic sections (each with 30 samples) by modeling the time elapsed between samples, $\Delta t$, and the stratigraphic height between samples, $\Delta h$, as gamma

distributions (Fig. 6b). This parameterization is a modification of the compound Poisson-Gamma chronology model (Haslett and Parnell, 2008). The shape parameter, $k$, of the gamma distribution controls whether the samples are unevenly (low $k$) or uniformly (high $k$) spaced. Different depositional histories are modeled by varying $k$ for the $\Delta t$ distribution between 0.1 and 10; stratigraphic completeness (i.e., the proportion of time preserved in the strata; Sadler, 1981) increases with $k$. To simulate regular stratigraphic spacing between samples, $\Delta h$ is modeled with $k = 100$ (Fig. 6c). The gamma distribution scale parameter

is fixed to $\theta = 1$ in all simulations. The age-height models are scaled to span the total time elapsed and total stratigraphic thickness of each section.

To isolate the effects of noisy proxy data on signal recovery, the proxy noise experiments are conducted using uniform depositional histories ($k = 100$ for both $\Delta t$ and $\Delta h$). Natural proxy variance and diagenesis are simulated by adding random

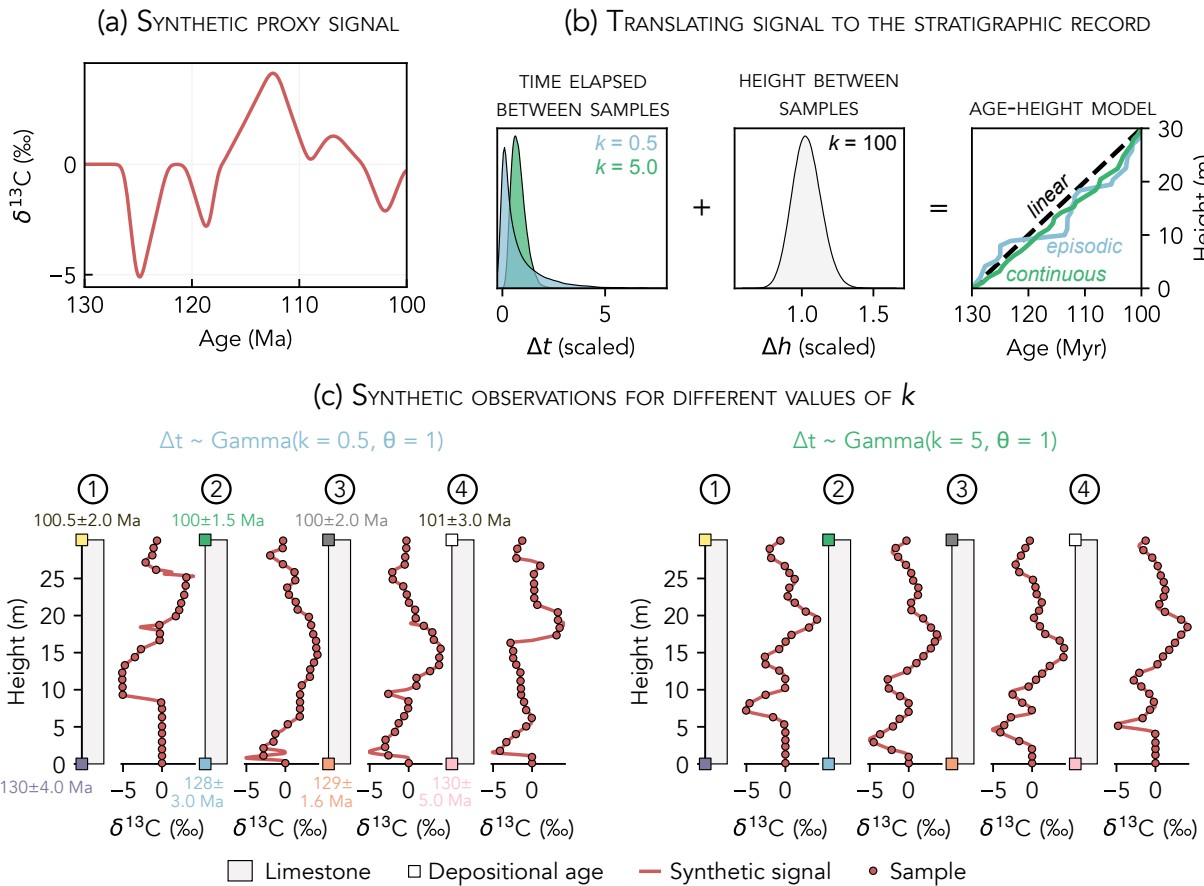

**Figure 6.** Procedure for generating the synthetic proxy data used in Sect. 3.3. The synthetic $\delta^{13}C$ signal in *(a)* is translated to the stratigraphic record by modeling the temporal ($\Delta t$) and stratigraphic ($\Delta h$) spacing between samples as gamma distributions *(b)*. The shape parameter, $k$, of the $\Delta t$ distribution controls whether sedimentation is episodic (samples unevenly spaced in time; low $k$) or continuous (samples evenly spaced in time; high $k$). Regular stratigraphic spacing is modeled using a $\Delta h$ distribution with $k$=100. The gamma distribtion scale parameter is fixed to $\theta$=1. The example sections in *(c)* were generated by modeling $\Delta t$ as a gamma distribution with $k = 0.5$ (left) or $k = 5$ (right).

Gaussian (white) noise to the $\delta^{13}C$ observations for each section. White noise is generated using a normal distribution with
$\mu = 0$ and $\sigma$ between $0.5$ (the low-noise endmember) and $5.0$ (the high-noise endmember).

### 3.1.2 Model parameters for experiments

**Table 1.** Prior distributions used for the model parameters in each experiment. RBF = radial basis function kernel; GP = Gaussian process.

| Experiment | RBF lengthscale | RBF variance | GP mean function | Geologic Noise ($\eta_{\text{section}}$) | Offset ($\phi$) |
|---|---|---|---|---|---|
| Single proxy (Sect. 3.2.1) | $\text{Wald}(\mu = 10, \lambda = 25) + 3$ | $\text{HalfNormal}(\sigma = 10)$ | $\text{Normal}(\mu = \mu_{data}, \sigma = 2\sigma_{data})$ | $\text{HalfCauchy}(\beta = 1)$ | per-section, $\text{Laplace}(\mu = 0, b = 2)$ |
| Multiproxy (Sect. 3.2.2) | $\text{Wald}(\mu = 10, \lambda = 25) + 3$ | $\text{HalfNormal}(\sigma = 10)$ | $\text{Normal}(\mu = \mu_{data}, \sigma = 2\sigma_{data})$ | $\text{HalfCauchy}(\beta = 1)$ | per-section, $\text{Laplace}(\mu = 0, b = 2)$ |
| Local bias, Experiment 1 (Sect. 3.2.3) | $\text{Wald}(\mu = 10, \lambda = 25) + 3$ | $\text{HalfNormal}(\sigma = 10)$ | $\text{Normal}(\mu = \mu_{data}, \sigma = 2\sigma_{data})$ | $\text{HalfCauchy}(\beta = 1)$ | per-environment, $\text{Laplace}(\mu = 0, b = 2)$ |
| Local bias, Experiment 2 (Sect. 3.2.3) | $\text{Wald}(\mu = 10, \lambda = 25) + 3$ | $\text{HalfNormal}(\sigma = 10)$ | $\text{Normal}(\mu = \mu_{data}, \sigma = 2\sigma_{data})$ | $\text{HalfCauchy}(\beta = 1)$ | per-section, $\text{Laplace}(\mu = 0, b = 2)$ |
| Proxy noise (Sect. 3.3.1) | $\text{Wald}(\mu = 4, \lambda = 15) + 2$ | $\text{HalfNormal}(\sigma = 10)$ | $\text{Normal}(\mu = \mu_{data}, \sigma = 2\sigma_{data})$ | $\text{HalfCauchy}(\beta = 1)$ | per-section, $\text{Laplace}(\mu = 0, b = 2)$ |
| Temporal noise (Sect. 3.3.2) | $\text{Wald}(\mu = 4, \lambda = 15) + 2$ | $\text{HalfNormal}(\sigma = 10)$ | $\text{Normal}(\mu = \mu_{data}, \sigma = 2\sigma_{data})$ | $\text{HalfCauchy}(\beta = 1)$ | per-section, $\text{Laplace}(\mu = 0, b = 2)$ |

Table 1 specifies the model parameter priors used for each synthetic experiment performed in Sect. 3.2 and 3.3. In all cases, the Gaussian process prior is specified such that it includes all geologically reasonable proxy signals given the age constraints and observed variance in the data. All experiments use the default priors for the per-section geologic noise ($\eta_{\text{section}}$) and offset ($\phi$) terms, which favor solutions with no local deviations from the common signal (Sect. 2.6.2).

### 3.1.3 Quantifying model performance

In the most basic sense, an inference is 'successful' if the synthetic proxy signal (i.e., the true proxy value during each time step) is captured by the posterior. Intuitively, we also know that inferences which capture the synthetic signal within narrow probability envelopes are superior to those that capture the synthetic signal within very wide probability envelopes. In other words, the quality of an inference is a function of both accuracy and precision. To capture this intuition, we evaluate model performance by calculating the average (across all time steps $\mathbf{t} = \{t_1, t_2, ... t_N\}$) likelihood of the synthetic proxy signal, $g(\mathbf{t})$, conditioned on the posterior distribution for the proxy value over time, $\theta_{f(\mathbf{t})}$:

$$P_{\theta_{f(\mathbf{t})}}(g(\mathbf{t})) = \frac{1}{N} \sum_{n=1}^{N} P_{\theta_{f(t_n)}}(g(t_n)) \tag{4}$$

For a given time step, the conditional likelihood of the synthetic proxy signal is high when the posterior proxy distribution is narrow and centered on the true proxy value (Fig. 7a). A low conditional likelihood indicates either that the posterior proxy distribution is wide, or that the posterior proxy distribution is narrow but assigns low probability to the true value (Fig. 7b). The

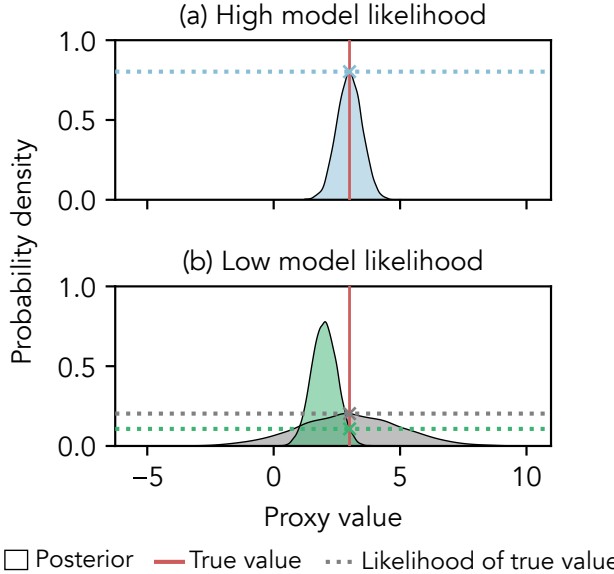

**Figure 7.** Calculating the conditional likelihood of the synthetic proxy signal during a single time step. In *(a)*, the conditional likelihood is high because the posterior proxy distribution is both narrow and centered on the true proxy value. In *(b)*, the conditional likelihood is low either because the posterior proxy distribution is wide (gray), or because it is narrow but offset from the true proxy value (green).

*mean signal likelihood* for a model, $P_{\theta_{f(\mathbf{t})}}(g(\mathbf{t}))$, is the average of the conditional likelihoods for all $N$ time steps. The mean signal likelihood can be used to compare performance among a group of candidate models associated with the same synthetic proxy signal.

## 3.2 Experiments: Testing the inference model

### 3.2.1 Single proxy inference

Our inference model is capable of accurately recovering signals recorded in synthetically generated stratigraphic data. We demonstrate this by applying the model to a slightly modified version of the observations in Fig. 5b. To simulate natural geochemical variability, zero-centered Gaussian noise with a standard deviation of $3.0$ (section 1), $0.75$ (sections 2, 3, 4, and 6), or $0.25$ (section 5) is added to the $\delta^{13}$C observations. To show how the model handles different types of age constraints, we assign additional depositional ages to sections 3 and 4 and a detrital age to section 1. Sections 3, 4, and 5 also host an unconformity-bounded glacial diamictite unit that serves as a correlative age constraint. Detrital zircons from the base of this diamictite unit have been dated at one location, providing a maximum age for overlying samples in all diamictite-bearing sections. The age for the top of the diamictite must be younger than the detrital zircon age and older than the oldest overlying depositional age constraint in all diamictite-bearing sections; this age range is modeled as a uniform distribution. When we apply our model to these synthetic data (Fig. 3.2.1a), the synthetic $\delta^{13}$C signal is captured fully by the $95\%$ envelope of the

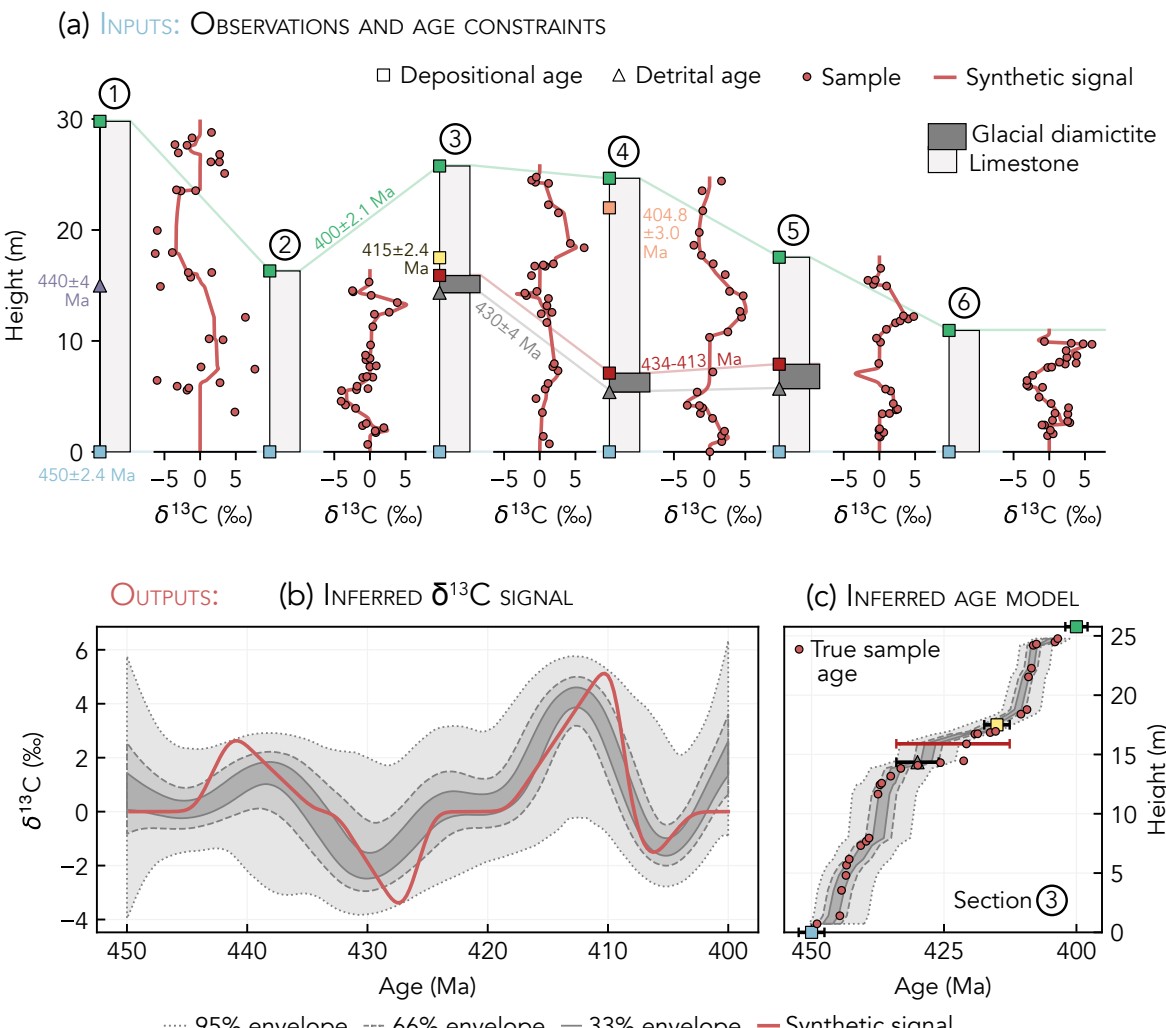

**Figure 8.** Testing the inference model using synthetic data. *(a)* Stratigraphic sections with age constraints and $\delta^{13}C$ observations. *(b)* $\delta^{13}C$ signal inference using the age constraints and $\delta^{13}C$ observations in *a*, with the synthetic $\delta^{13}C$ signal plotted for comparison. The 95% envelope marks the 2.5th and 97.5th percentiles of the posterior $\delta^{13}C$ distribution for each time step, the 66% envelope marks the 17th and 83rd percentiles, and the 33% envelope marks the 33.5th and 66.5th percentiles. *(c)* Posterior age model for section 3, with samples plotted by their true age.

inference, and mostly (80% of the time) falls within the 66% envelope of the inference (Fig. 8b). The true sample ages also typically fall within the 95% envelopes of the posterior section age models (Fig. 8c).

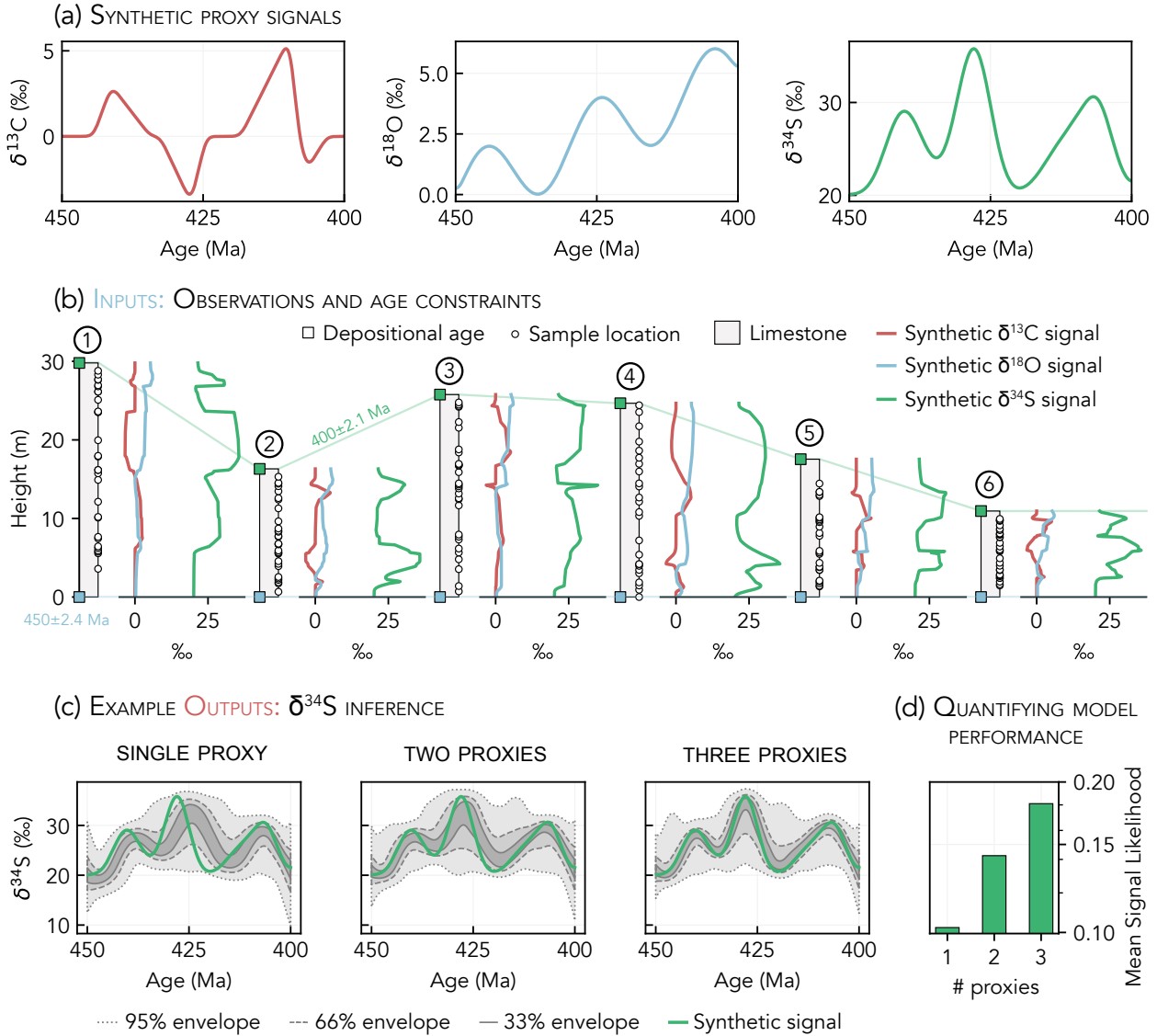

**Figure 9.** Synthetic example of multiproxy inference. *(a)* Synthetic proxy signals for $\delta^{13}$C (as in Fig. 5a), $\delta^{18}$O, and $\delta^{34}$S. *(b)* Proxy signals in *a* mapped to each stratigraphic section. Sample heights are marked on lithostratigraphic columns; the additional 'geologic noise' added to the proxy observations is not illustrated. *(c)* Inferred $\delta^{34}$S signal for models that include one, two, or three proxies. The two-proxy inference is plotted using the combined posteriors for the $\delta^{34}$S-$\delta^{13}$C and $\delta^{34}$S-$\delta^{18}$O inferences. The inferred timing and magnitude of each excursion more closely matches the synthetic signal as additional proxies are considered. *(d)* Mean $\delta^{34}$S signal recovery (Eq. 4) for the one-, two-, and three-proxy models, where the two-proxy signal likelihoods are calculated using the combined posteriors for the $\delta^{34}$S-$\delta^{13}$C and $\delta^{34}$S-$\delta^{18}$O inferences. Note the logarithmic scale of the y-axis. Recovery of the synthetic proxy signal improves as the number of proxies increases.

### 3.2.2 Multiproxy inference

Next, we demonstrate how simultaneously inferring signals for multiple proxies ('multiproxy inference') can improve signal recovery. To build a multiproxy data set, we first generate synthetic $\delta^{18}$O and $\delta^{34}$S signals that span the same time interval as the synthetic $\delta^{13}$C signal (Fig. 9a). Then, we translate these synthetic signals to the stratigraphic record following the same procedure used to produce the $\delta^{13}$C observations (Fig. 5). Fig. 9b illustrates how all three proxy signals are mapped to the synthetic stratigraphic sections. White noise with an amplitude of 0.75‰, 0.75‰, and 1.5‰ is added to the $\delta^{13}$C, $\delta^{18}$O, and $\delta^{34}$S observations, respectively. All proxy observations are modeled with measurement uncertainties of 0.1‰. To isolate the effects of additional proxy data on the signal inference, only minimum and maximum depositional age constraints are assigned to each section.

Reconstruction accuracy improves for all metrics as the inference model considers additional proxies. Each added proxy contributes new information for the model to learn from, which can help to better constrain age models for sections where other proxy records were relatively uninformative. Considering more proxies in concert will, on average, create more accurate and well-constrained age models, which leads to better signal recovery (assuming the proxy data are not significantly biased). To demonstrate, Fig. 9c shows the inferred $\delta^{34}$S signal for one-, two-, and three-proxy inferences. As additional proxies are considered, the 33% envelope of the inference more accurately approximates the synthetic signal. For instance, the largest positive $\delta^{34}$S excursion has a peak value of 35.8‰ at 428 Ma. The single-proxy inference underestimates the peak value and timing of the excursion, with a most likely maximum of 32.8‰ occurring at 424 Ma. Both of these estimates improve when two proxies are considered, with a most likely excursion maximum of 33.3‰ occurring at 427 Ma. The three-proxy inference yields the most accurate reconstruction of the excursion, with a most likely peak $\delta^{34}$S value of 34.4‰ at 428 Ma. Quantified signal recovery (Eq. 4) for these experiments confirms that model performance improves as additional proxies are considered (Fig. 9d); for the $\delta^{34}$S inferences in Fig. 9c, the mean signal likelihoods for models that include one, two, and three proxies are 0.10, 0.14, and 0.18, respectively.

### 3.2.3 Local environmental bias

Carbonate sediments formed in different depositional environments may have average $\delta^{13}$C values that are depleted or elevated with respect to contemporaneous open-ocean DIC (e.g., Patterson and Walter, 1994; Geyman and Maloof, 2021; Pederson et al., 2021; Trower et al., 2024). To test how the inference model handles the local environmental bias that can exist in real data, we alter the synthetic data (Fig. 5b) in two ways. First, we construct a new lithostratigraphic column for each section by taking a cross-section through a synthetic carbonate platform (Fig. 10a). At each section location, stratigraphic changes in environment represent the lateral migration of adjacent depositional environments. Using these new lithostratigraphies, we then modify the proxy data such that the $\delta^{13}$C of samples from each depositional environment is variably offset from the global signal (Fig. 10b-c). The offset for each depositional environment is modeled as a normal distribution centered on the average difference between local and global $\delta^{13}$C$_{\text{DIC}}$ (Fig. 10b). These offsets, while schematic, are broadly consistent with modern observations. For example, subtidal/lagoonal sediments often have slightly elevated $\delta^{13}$C due to photosynthetic activity in restricted banktop

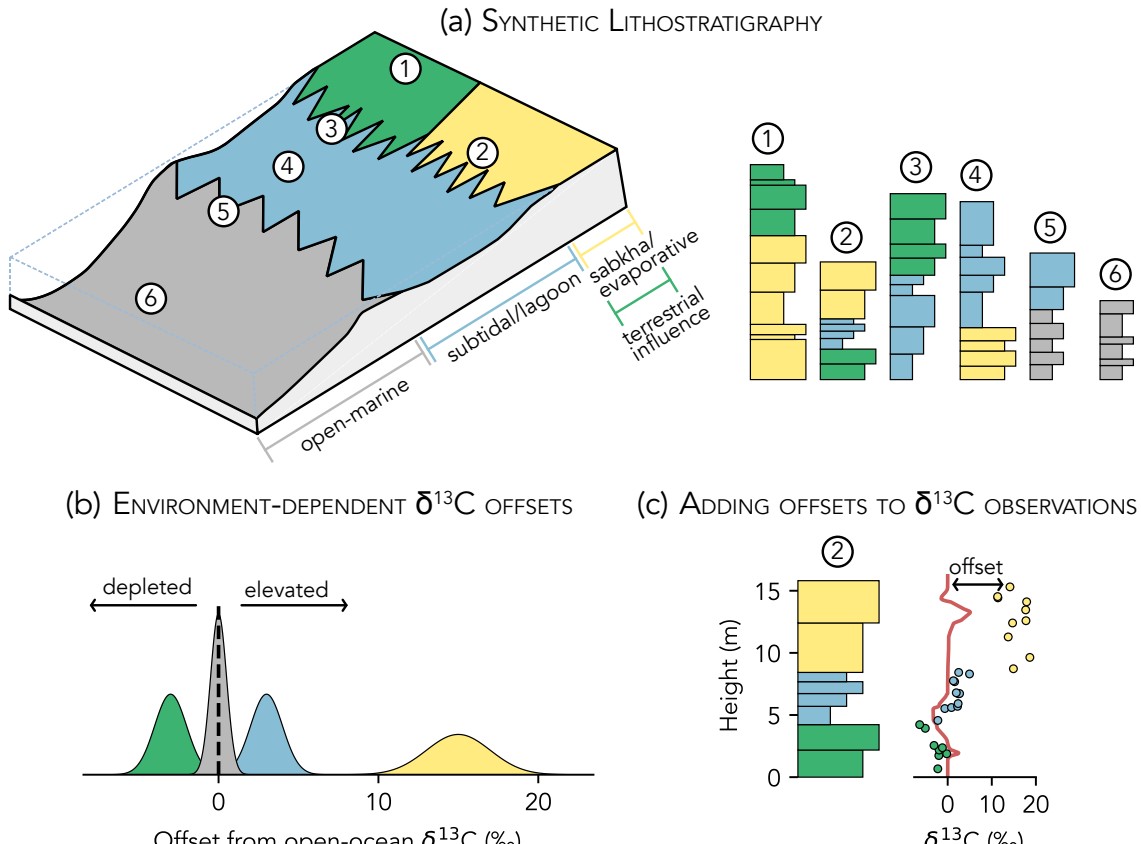

**Figure 10.** Modifying the synthetic data to simulate local environmental bias. *(a)* Synthetic carbonate platform and lithostratigraphic columns. *(b)* Distribution of $\delta^{13}C$ offsets for each depositional environment. Subtidal and sabkha environments are elevated with respect to open-ocean $\delta^{13}C_{DIC}$, while terrestrially-influenced environments are depleted. *(c)* The $\delta^{13}C$ value for each sample is offset from the synthetic signal by a value drawn from the appropriate probability distribution in *b*.

waters (Geyman and Maloof, 2019), while terrestrially-influenced environments (e.g., mangrove ponds) can have depleted $\delta^{13}C$ due to input of groundwater charged with DIC derived from remineralized organic matter (Patterson and Walter, 1994). The modified stratigraphic sections and proxy observations are in Fig. 11a.

When we build geological context into the model by assigning a unique offset term to samples from each environment (i.e., all subtidal samples are shifted by $\phi_{\text{subtidal}}$ relative to the common signal), the synthetic signal is captured by the $95\%$ envelope of the $\delta^{13}C$ inference (Fig. 11b). The model also correctly infers the relative offsets between each depositional environment (Fig. 11c). However, the inferred proxy signal is elevated by a mean value of $1.4‰$ (averaged across all time step) relative to the synthetic signal, while the mean value of each offset term is underestimated. Specifically, the mean offsets for open-marine,

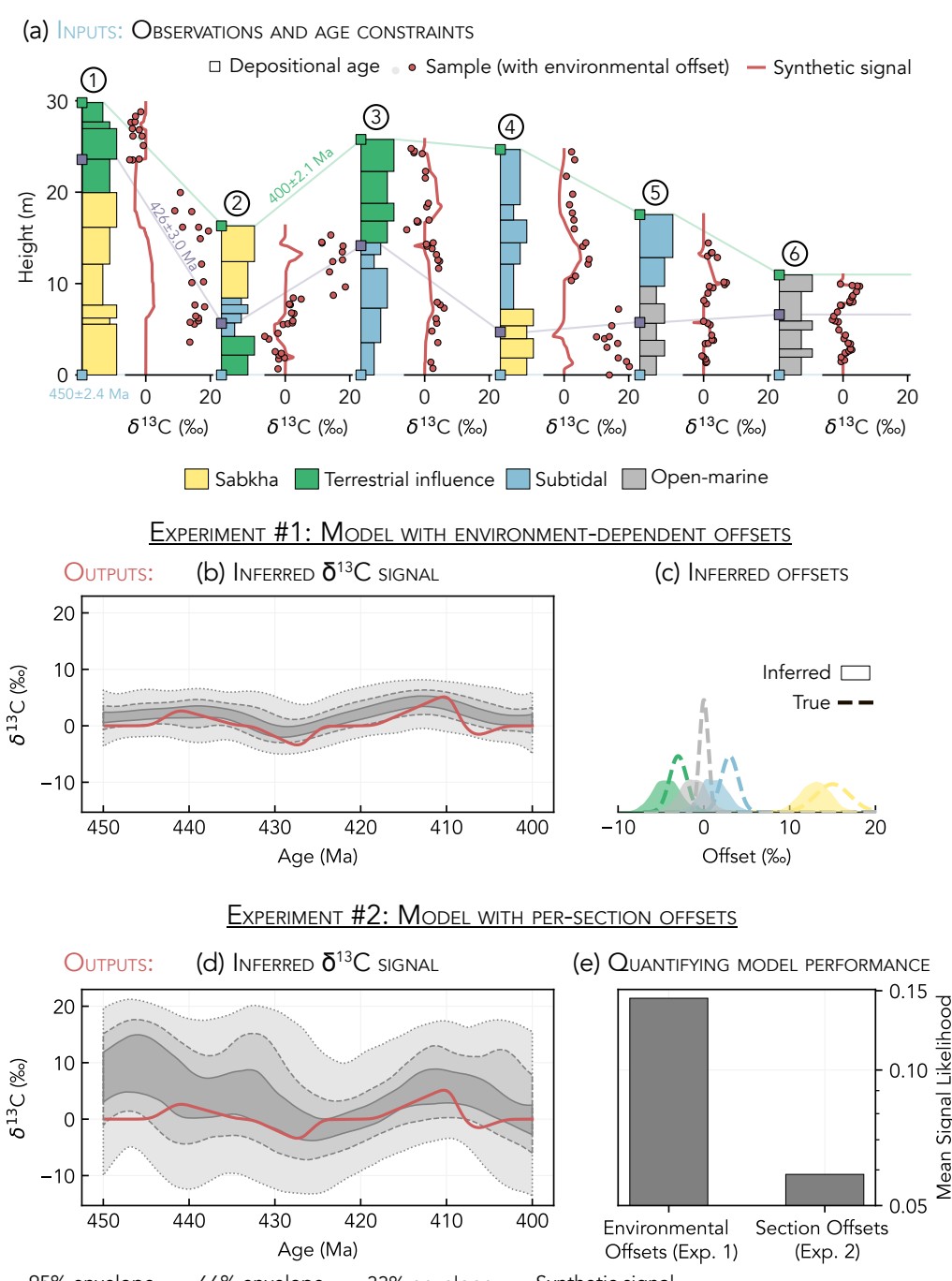

**Figure 11.** Data and results for local bias experiments. *(a)* Synthetic stratigraphic sections and $\delta^{13}C$ observations. Environment-dependent offsets have been added to the $\delta^{13}C$ data as described in Fig. 10. *(b)* $\delta^{13}C$ signal inference when all samples from a given depositional environment share an offset term in the model; the synthetic signal is plotted for comparison. *(c)* Inferred offsets for each environment, with the true offsets (as in Fig. 10b) plotted for comparison. *(d)* $\delta^{13}C$ signal inference when all samples from the same section share an offset term. *(e)* Mean signal likelihoods (Eq. 4) for the $\delta^{13}C$ inferences in *b* (Experiment 1) versus *d* (Experiment 2). Note the logarithmic scale of the y-axis.

355 subtidal, terretrially-influenced, and sabkha environments are underestimated by $1.2‰$, $1.8‰$, $1.4‰$, and $2.0‰$, respectively. Both of these deviations occur because the $\delta^{13}C$ observations have an overall positive bias relative to the synthetic signal.

 If we lacked the geologic information required to group samples by depositional environment, we instead might assume that each location represents a unique environment and assign a unique offset term to each section (i.e., all samples in section 1 are shifted by $\phi_1$ relative to the common signal). As a result of this less informed modeling choice, the $\delta^{13}C$ signal inference must

360 have wider probability envelopes in order to capture all of the observations (Fig. 11d). While the synthetic signal still falls within the $95\%$ envelope of the inference, using per-section offset terms leads, on average, to a $3.3‰$ overestimation of $\delta^{13}C$. To quantify the value added by considering paleoenvironmental context, we calculate that the mean signal likelihood (Eq. 4) for the model with environment-dependent offsets is 0.14, compared to 0.06 for the model with per-section offsets (Fig. 11e). This experiment demonstrates that the accuracy of signal reconstructions may be improved by conducting careful geologic

365 work to categorize samples by depositional environment.

### 3.3 Experiments: Recovering signals from noisy data

Real data are almost always influenced by 'geologic noise' that can hinder recovery of primary geochemical signals. We consider two general categories of geologic noise: 'proxy noise', which changes proxy values relative to the common signal, and 'temporal noise', which refers to irregularity in the relationship between stratigraphic height and time resulting from

370 episodic sedimentation. Here, we leverage synthetic experiments to 1) evaluate the effects of geologic noise on proxy signal recovery, and 2) consider how different types of geologic noise might be recognized in the rock record using our modeling framework.

#### 3.3.1 Noise in proxy data

We simulate diagenesis and natural proxy variance by adding random (white) noise with amplitude ($\sigma_{noise}$) between $0.5$ and

375 $5.0$ to the proxy data (Sect. 3.1.1). For each experiment, we apply the inference model to four sections with added proxy noise generated using the same $\sigma_{noise}$ value. To ensure that our findings are generalizable, the reported results for each experiment represent the average of three trials, where each trial is executed using different randomly generated noise. Example proxy observations are shown in Fig. 12a.

 As the amplitude of added white noise increases, the model is only able to resolve higher-amplitude and longer-term (lower-

380 frequency) features of the proxy signal. The confidence envelopes of the signal inference become wider and smoother with increasing $\sigma_{noise}$, effectively 'blurring' the signal as progressively larger isotopic excursions are obscured (Fig. 12b). In this example, signal recovery initially declines as $\sigma_{noise}$ increases and stabilizes above $\sigma_{noise} = 2$ (Fig. 12d). These experiments suggest that signal recovery deteriorates dramatically when the amplitude ($2\sigma$) of random noise meets or exceeds the amplitude of the common proxy signal. In general, however, the model's resolving power is sensitive to both the amplitude of added noise

385 and the density of geochronological age constraints, where more age constraints may help to combat the blurring effect of proxy noise.

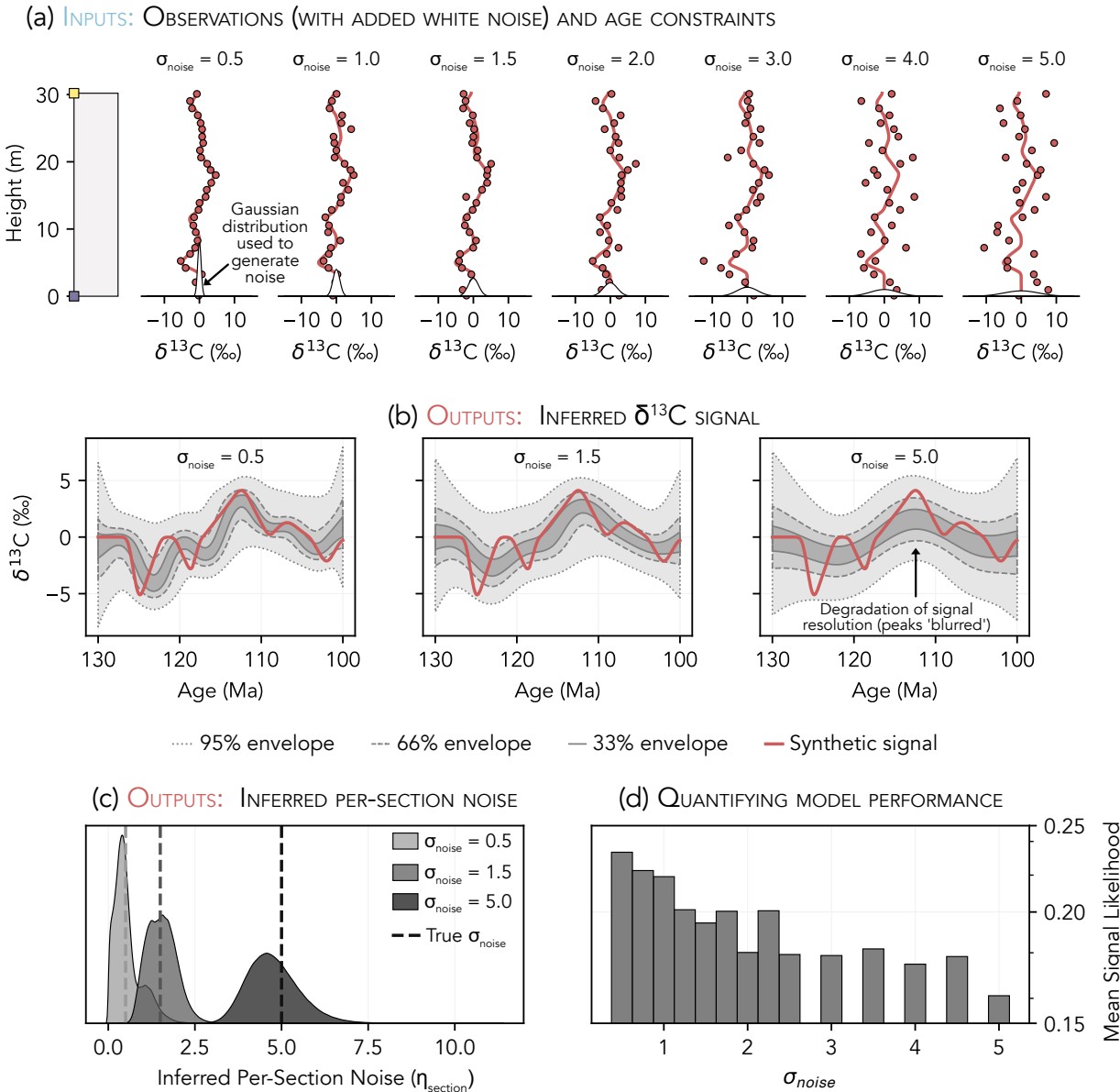

**Figure 12.** Data and results for proxy noise experiments. *(a)* Example of a synthetic stratigraphic section with noisy proxy observations. The proxy observations in each panel were produced by adding white noise generated from a zero-centered normal distribution with standard deviation $\sigma_{\text{noise}}$ (see probability distributions plotted on the x-axes). *(b)* $\delta^{13}C$ signal inferences for experiments run with four stratigraphic sections and $\sigma_{\text{noise}}$ values of 0.5, 1.5, and 5.0; each inference is plotted using the combined posteriors for three trials. *(c)* Inferred per-section geologic noise terms ($\eta_{\text{section}}$) corresponding to the $\delta^{13}C$ signal inferences in *b*. Each distribution reflects the combined posteriors for all sections and trials. For comparison, the amplitude of white noise added to the proxy observations ($\sigma_{\text{noise}}$) for each experiment is plotted as a vertical dashed line. *(d)* Mean signal likelihoods (Eq. 4) for models run using proxy observations generated with different $\sigma_{\text{noise}}$ values. Note the logarithmic scale of the y-axis. Higher mean signal likelihoods correspond to better recovery of the synthetic proxy signal.

The resolving power of the model sometimes can be improved by removing particularly noisy sections from the inference. Noisy sections can be identified using the posterior distributions for the per-section geologic noise terms ($\eta_{section}$; Sect. 2.6.2), which accurately capture the amplitude of added white noise in our experiments. For example, the median inferred $\eta_{section}$ value is 0.4, 1.5, and 4.7 (averaged across all sections and trials) for data generated with $\sigma_{noise}$ values of 0.5, 1.5, and 5.0, respectively (Fig. 12c).

### 3.3.2 Non-uniform depositional histories

Deep-water environments with relatively constant sedimentation rates classically are considered to be the most reliable archives of past proxy change. However, since a large fraction of deep-sea sediments older than ∼200 Ma have been subducted at continental margins, reconstructions of marine proxy signals prior to the mid-Mesozoic instead rely primarily on sediments deposited in shallow-water environments, where more episodic deposition leads to low stratigraphic completeness. Here, we use our model to consider how these shallow-water reconstructions may stack up to those based on more complete deep-sea records. We build stratigraphic sections with depositional histories ranging from episodic ($k = 0.1$) to uniform ($k = 10$) following the procedure detailed in Sect. 3.1.1. For each experiment, we apply our inference model to between four and ten sections generated using the same $k$ value.

Our model validates the intuition that continuous deep-sea records are preferable to less complete shallow-water records. For a fixed number of stratigraphic sections, signal recovery generally improves as $k$ increases (i.e., as sedimentation becomes less episodic, increasing completeness) (Fig. 13c). For each $k$ value, signal recovery also improves as additional sections are considered. Still, all hope is not lost for stratigraphers working in ancient shallow-water strata: signal recovery for models that include a large number of highly incomplete (low-$k$) sections is comparable to signal recovery for models with a lower number of complete (high-$k$) sections (Fig. 13b). In other words, quantity can compensate for quality. For example, the mean signal likelihood for the 2-section model with $k = 10$ is 0.20, compared to 0.14 for the 2-section model with $k = 0.5$ and 0.21 for the 10-section model with $k = 0.5$. As deposition becomes increasingly episodic (lower $k$), a greater number of sections is required to achieve comparable performance.

## 4 Discussion

### 4.1 Diagnosing non-global signals

The case studies in Sect. 3 collectively illustrate that our inference model can reconstruct proxy signals over time using time-uncertain, biased, noisy, and incomplete stratigraphic data. However, real-world data are particularly complex, and it is important to carefully examine the inference results before interpreting the reconstructed proxy signal. In this section, we discuss how to analyze the posterior to better understand the range of global and non-global processes influencing the results.

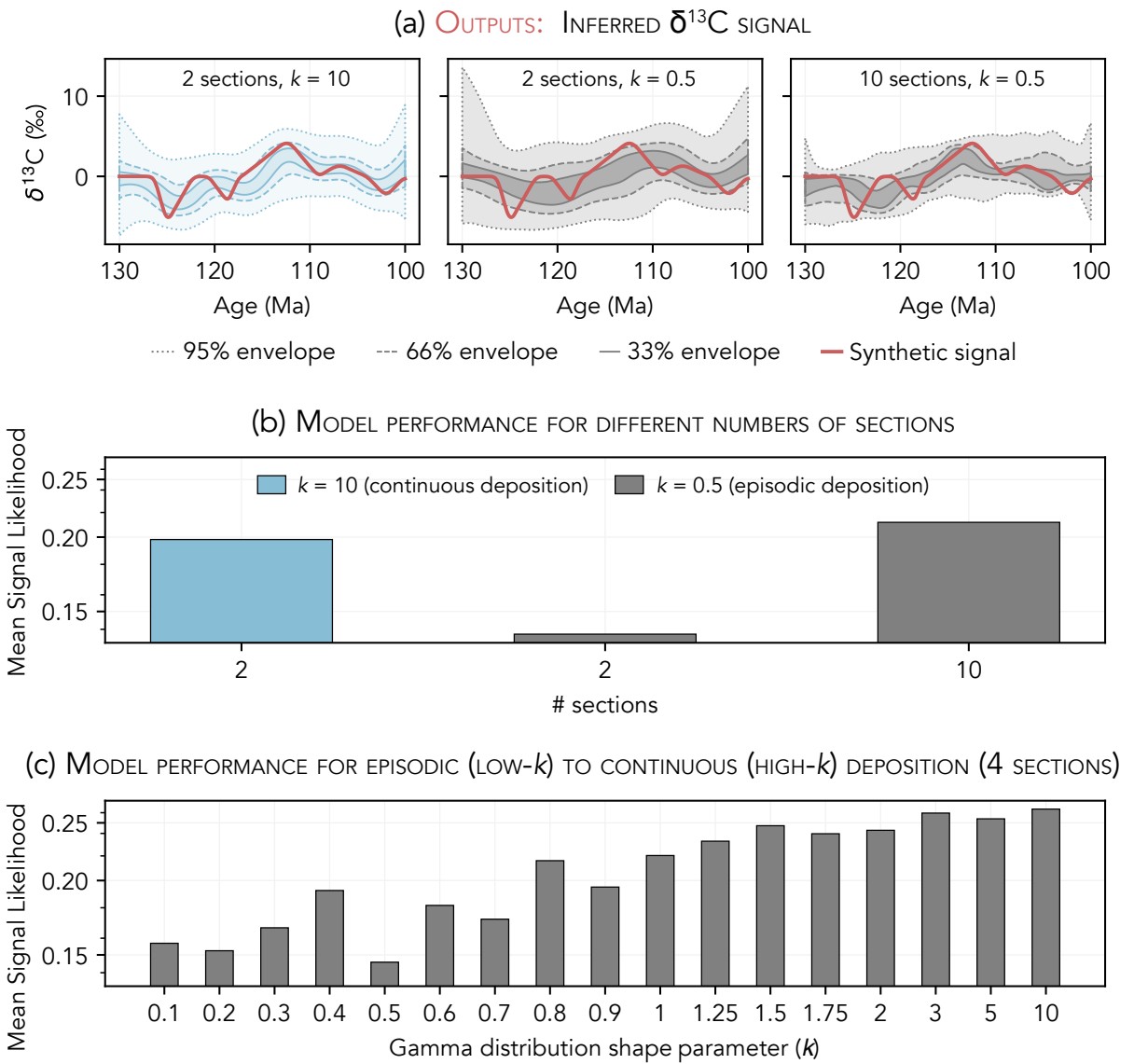

**Figure 13.** Results of temporal noise experiments. *(a)* Example $\delta^{13}$C signal inferences for models that include two sections with $k = 10$ (left), two sections with $k = 0.5$ (center), and ten sections with $k = 0.5$ (right). *(b)* Mean signal likelihoods (Eq. 4) for the signal inferences in *a*. *(c)* Mean signal likelihoods for models run with four sections and different $k$ values. Note the logarithmic y-axis scale in *b* and *c*.

### 4.1.1 Reconstructing local environmental bias

A number of marine geochemical proxies are sensitive to local processes that can increase or decrease proxy values relative to the global average. For example, carbonate $\delta^{13}$C is influenced by local biological activity (Geyman and Maloof, 2019; Pederson

et al., 2021), groundwater discharge (Patterson and Walter, 1994), and the abundance of different grain types with distinct geochemical fingerprints (Gischler et al., 2009; Geyman and Maloof, 2021). Similarly, carbonate $\delta^{18}O$ is a complex function of temperature (Romanek et al., 1992), local hydrology (LeGrande and Schmidt, 2006), and global ice volume (Shackleton, 1967). Considering other types of sediments, recent work indicates that pyrite $\delta^{34}S$ is sensitive to sedimentation rate (Pasquier et al., 2021; Li et al., 2024), which varies dramatically between shallow and deep-water environments, while spatial patterns in sedimentary $\delta^{15}N$ reflect gradients in nutrient availability and redox conditions (Mollier-Vogel et al., 2012; Motomura et al., 2024).

When the direction and/or magnitude of local bias varies among different depositional environments in the same basin, the lateral migration of adjacent environments as sediment accumulates can generate stratigraphic changes in proxy values that are potentially unrelated to large-scale biogeochemical cycling (e.g., Holmden et al., 1998; Geyman and Maloof, 2021). These spurious environment-driven signals both complicate correlation among sections and obscure the true nature of proxy change over time. We can use the inference model to distinguish between local bias and temporal proxy perturbations by assigning a unique *offset* ($\phi$) term to samples from different environments (for guidance on choosing an appropriate offset parameterization, see Sect. 4.3.4). Learning from the proxy observations, the model simultaneously isolates the common signal recorded by all sections and estimates the magnitude and direction of the bias characterizing each environment. While reconstructing global proxy change over time often is our main objective, these quantitative estimates of local bias encode independently valuable information about local paleoenvironment.

The inferred offsets for different environments can be used to pose testable hypotheses about the local processes influencing proxy values within a given basin. To demonstrate, let us consider how the results of Experiment 1 in Sect. 3.2.3 (Fig. 11b-c) might be interpreted in the real world. Recall that the model infers non-zero offsets because the stratigraphic proxy trends in all sections covary (i.e., each section is influenced by the same global signal), but the absolute proxy values within each environment are shifted. In our example, non-zero offsets indicate that local influences on $\delta^{13}C$ outpace the processes by which shallow waters chemically equilibrate with the global ocean (physical mixing and air-sea gas exchange), which suggests communication between platform-top waters and the open ocean is physically restricted. The magnitude and direction of the offsets provide clues about which processes are operating in each environment. For example, terrestrially-influenced samples are depleted by $3.1 \pm 1.2‰$ relative to open-marine samples, which indicates the DIC pool is dominated by isotopically light carbon derived from an outside source (e.g., groundwater carrying soil-derived carbon; Patterson and Walter, 1994). Conversely, the $14.2 \pm 1.6‰$ elevation in $\delta^{13}C$ in evaporative sabkha environments is consistent either with extreme photosynthetic enrichment of $\delta^{13}C$ in restricted waters (which requires high sedimentary organic matter content), or more likely with active methanogenesis (e.g., Cadeau et al., 2020), which strongly fractionates carbon by preferentially sequestering $^{12}C$ in methane ($CH_4$).

The above analysis serves as a guide for interpreting analogous observations from the rock record. For example, many previous workers have interpreted differences in $\delta^{13}C$ profiles within and between basins as evidence of persistent $\delta^{13}C_{DIC}$ gradients in Earth's ancient oceans. In the Precambrian record, Prave et al. (2022) suggest that the Paleoproterozoic Lomagundi-Jatuli excursion – canonically interpreted as reflecting global carbon cycling during the Great Oxidation Event (e.g., Karhu and

Holland, 1996) – instead may largely be an artifact of local environmental conditions. During the Paleozoic, Schiffbauer et al. (2017) propose that the Cambrian Steptoean positive carbon isotope excursion (SPICE) is most parsimoniously explained as the migration of a water-depth $\delta^{13}C_{DIC}$ gradient during sea level rise, while Yang et al. (2024) similarly postulate that the Ordovician Hirnantian carbon isotope excursion (HICE) partly reflects $\delta^{13}C_{DIC}$ stratification during sea level fall associated with glaciation. Our modeling framework can be used to directly test these hypotheses and to quantify the probable magnitude and structure of ancient $\delta^{13}C_{DIC}$ (and other geochemical proxy) gradients. By considering many reconstructions in tandem, we also can gain more general insight to how the operation of different biogeochemical cycles and sedimentary systems has coevolved with life and climate over Earth history.

### 4.1.2   Fingerprinting sources of geologic noise

In the previous section, we described how consistent biases between preserved proxy values and the common signal are captured by the offset term ($\phi$) in the model. Any misfits that cannot be modeled as constant offsets instead must be accounted for by the inferred per-section geologic noise term ($\eta_{section}$). This geologic noise term encapsulates non-global processes, such as local biogeochemical cycling and diagenesis, that act to decouple preserved proxy values from the global average. Here, we discuss how the model posterior can be used to diagnose which processes are responsible for this decoupling in sections with high inferred $\eta_{section}$ terms. In Sect. 4.3.3, we provide advice on how these high-noise sections should be handled.

Stratigraphic patterns in the residuals between the proxy observations for each section and the inferred proxy signal can help to fingerprint the source of geologic noise. For instance, the extent to which proxy values are altered during meteoric diagenesis is predicted to increase with proximity to upsection subaerial exposure surfaces (Allan and Matthews, 1982; Dyer et al., 2017). Therefore, meteoric diagenesis may be indicated if the residuals increase as an exposure surface is approached. On the other hand, unstructured (i.e., randomly distributed) residuals are potentially consistent with burial diagenesis or mixing between primary and authigenic phases, which could either increase the variance of or homogenize proxy values (Frauenstein et al., 2009; Metzger and Fike, 2013; Martindale et al., 2015; Ahm et al., 2019). This scenario is analogous to our white noise experiments (Sect. 3.3.1). However, diagenesis cannot be definitively diagnosed based on patterns in the residuals alone, because primary geochemical variability can confer similar trends. For instance, in modern environments bulk carbonate $\delta^{13}C$ exhibits spatial variability of up to $\sim 5‰$ (Weber, 1967; Weber and Woodhead, 1969; Gischler et al., 2009; Swart et al., 2009; Geyman and Maloof, 2021), while the $\delta^{13}C$ of different grain types can vary by $\sim 10‰$ (Lowenstam and Epstein, 1957; Geyman and Maloof, 2021). This variability may appear as random noise (unstructured residuals), but also could impart stratigraphic trends in the residuals if proxy values covary with sedimentary facies (i.e., the type and size distribution of constituent grains). To more definitely distinguish between primary variability and diagenesis, we suggest checking for correlations between the residuals and independent diagenetic indicators (e.g., trace element concentrations; Brand and Veizer, 1980).

Multiproxy inference may provide additional insight to both the processes responsible for high posterior geologic noise and whether the inferred proxy signal might be biased by non-global processes. If a given section has high posterior geologic noise terms for multiple proxies that are susceptible to diagenetic alteration (e.g., $\delta^{13}C$, $\delta^{18}O$, and $\delta^{44/40}Ca$), then diagenesis is the likely culprit. However, if only one proxy is 'noisy', while other proxies that are similarly (or more) susceptible to diagenesis

are not, then primary variability within the depositional environment may be the more likely cause. In addition, synchronous inferred changes in proxy systems with disparate residence times, or in systems that are not expected to exhibit secular change (e.g., some trace element concentrations), may indicate that the common signal has been biased by diagenesis. For example, globally synchronous meteoric alteration associated with eustatic sea level fall might impart similar stratigraphic trends and covariance in carbonate $\delta^{13}C$, $\delta^{18}O$, $\delta^{44/40}Ca$, and Sr/Ca in many different locations. Consequently, the model may spuriously infer coincident temporal changes in the proxy values for each system that are unrelated to temporal changes in seawater chemistry. We elaborate on possible non-global influences on the common proxy signal in Sect. 4.5.

## 4.2 Comparison with alternative approaches

Most existing chemostratigraphy algorithms aim to objectively and reproducibly correlate stratigraphic sections; composite records of proxy change over time then are constructed only after the correlation step is complete. In contrast, our model explicitly seeks to reconstruct past changes in large-scale biogeochemical cycling by inferring the common, but potentially unobserved or 'hidden' (i.e., not directly preserved at any location), proxy signal recorded by all stratigraphic sections. Correlation and age model construction are integral parts of the signal reconstruction process, but are not the main objective. This distinction is important because by explicitly inferring the common proxy signal, rather than simply optimizing the alignment among sections, our model distinguishes between observed stratigraphic changes in proxy values – which may be influenced by a host of non-global processes – and temporal changes in global proxy values. Aside from this big-picture distinction, there are several more subtle, albeit significant, differences between our model and other chemostratigraphy algorithms; we expand on these differences in the following paragraphs.

The majority of existing quantitative algorithms for stratigraphic correlation rely on dynamic time warping, which is a deterministic least-squares approach for finding the optimal alignment between two sections (Lisiecki and Lisiecki, 2002; Hay et al., 2019; Hagen et al., 2024). Compared to more subjective manual approaches, these algorithms constitute a significant step toward a more objective and reproducible approach to correlation. Recent applications of dynamic time warping have yielded important insights to Ediacaran, Cambrian, and Paleogene $\delta^{13}C$ records (Hay et al., 2019; Ajayi et al., 2020; Hagen and Creveling, 2024), as well as many other times and proxy systems (Peti et al., 2020; Hagen and Harper, 2023; Reilly et al., 2023; Lilkendey et al., 2025). Our model builds on these previous efforts by addressing several inherent shortcomings of least-squares algorithms that limit their effectiveness when considering ancient shallow-water strata.

Our model has three significant advantages over least-squares correlation algorithms. First, existing algorithms aim only to find the optimal alignment of proxy data between sections, and generally consider geochronological, biostratigraphic, and lithostratigraphic age constraints only after correlations have been made (Hay et al., 2019; Hagen and Creveling, 2024). Absolute age models then are constructed using independent modeling tools (e.g., Johnstone et al., 2019; Trayler et al., 2020; Zhang et al., 2023). Our model integrates correlation and age model construction by enforcing absolute and relative age constraints during the alignment step. The resulting age model for each section incorporates both uncertainty that results from interpolating between geochronological ages (typical for Bayesian age models) and uncertainty in the alignment among sections. Excepting the algorithm developed by Bloem and Curtis (2024), which leverages a computational model of sediment accumulation to

correlate sections within a single basin, other existing algorithms do not explicitly quantify alignment uncertainty. Although dynamic time warping can be leveraged to explore different plausible alignments between pairs of sections, typically only the 'best' alignment is used to construct composite proxy records (Hagen and Creveling, 2024), likely due to the computational expense of accounting for all alignment permutations when considering more than a few sections. Second, most algorithms require the user to choose one section to act as the 'backbone' for correlation (Lisiecki and Lisiecki, 2002; Hay et al., 2019; Hagen et al., 2024). This choice may bias inferences about the true nature of proxy change over time, especially if the chosen 'backbone' is incomplete (e.g., shallow-water sections where deposition is episodic) or locally biased. In contrast, our model simultaneously aligns all sections while accounting for both local offsets (Fig. 11) and random proxy 'noise' related to diagenesis or local variability (Fig. 12). Third, no existing algorithm allows for simultaneous correlation of stratigraphic data for multiple geochemical proxies, which can improve signal reconstructions by narrowing the range of plausible alignments among sections (Fig. 9). While our experiments focus on correlation of geochemical proxies, our model can be applied to any quantitative stratigraphic data (e.g., relative grain size measurements, paleomagnetic data, gamma ray logs).

Although our model generally is better equipped to deal with observations from ancient shallow-water environments, these improvements come at the expense of speed and computational complexity (Sect. 4.4). For problems involving relatively continuous sections that are unlikely to be locally biased, a least-squares approach may be preferable. Dynamic time warping also provides a fast and visually simple means of exploring different plausible alignments between sections. While our model also catalogues different plausible alignments, visualizing these possibilities requires more targeted interrogation of the posterior age models for each section. As such, dynamic time warping may provide a more accessible first-order exploration of different solutions. However, quantifying and propagating the uncertainties stemming from these different solutions (when e.g., constructing composite proxy records) is more challenging in a least-squares framework.

Currently, the only comparable Bayesian chemostratigraphy algorithm is the BIGMACS model developed by Lee et al. (2023). BIGMACS is similar to StratMC in that it both correlates geochemical proxy profiles and constructs section age models. However, unlike StratMC, its age modeling approach relies on prior information about sedimentation rates in deep-sea cores that generally is not available for more ancient and incomplete shallow-water stratigraphies. BIGMACS also does not allow for simultaneous correlation of multiple geochemical proxies (e.g., $\delta^{18}$O and $\delta^{13}$C). BIGMACS does include per-section proxy offsets, and it uses Gaussian process regression to construct a composite proxy 'stack'. Importantly, the stack construction and correlation/age modeling steps are performed iteratively rather than simultaneously, meaning that sections are aligned to the current stack – essentially the mean and variance of the data over time, excluding outliers – rather than to a common 'hidden' proxy signal. BIGMACS also requires the user to provide an initial alignment target, necessitating some prior knowledge of the signal to be reconstructed.

### 4.3 Improving signal recovery

Proxy signal recovery can be improved by collecting additional data or discarding low-quality data with proper justification. The following subsections describe how to analyze the posterior results to guide real-world data collection efforts and determine which existing data should be more closely scrutinized before including them in future analyses.

### 4.3.1 Identifying 'gaps' in the proxy data

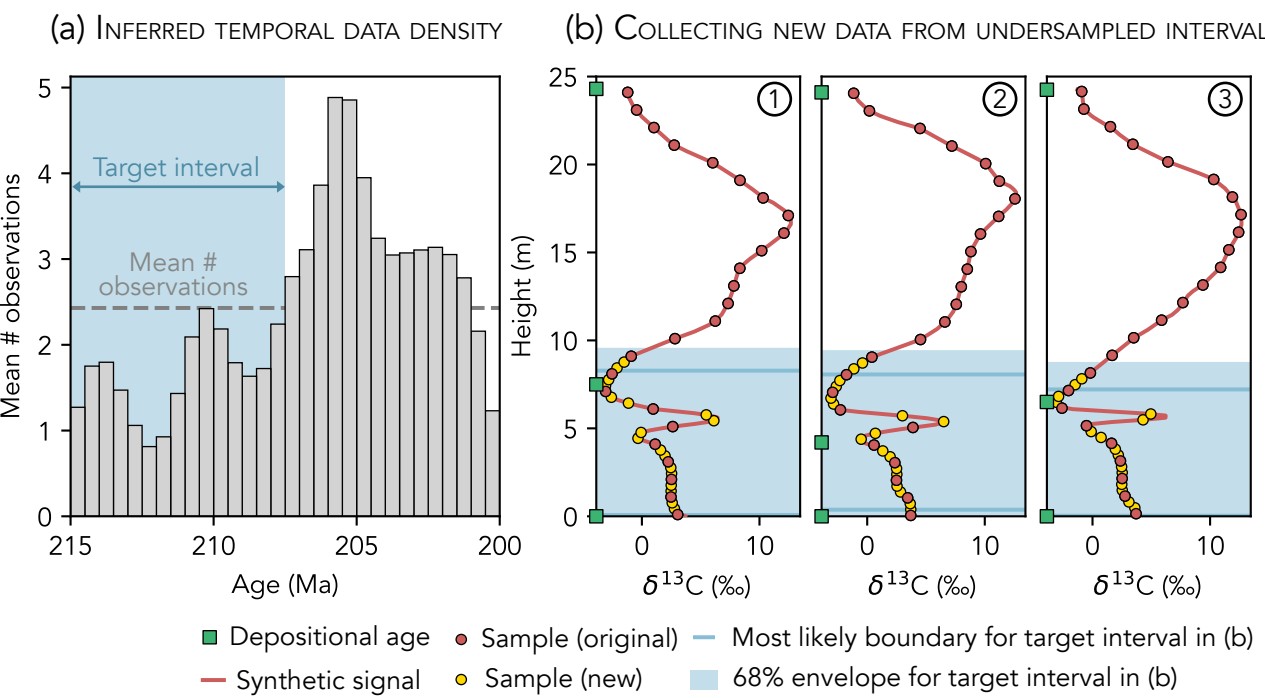

**Figure 14.** Workflow for using the model outputs to identify gaps in the proxy data. *(a)* Inferred temporal density of $\delta^{13}$C observations in 0.5 Myr bins. The blue shaded region from 215 to 207.5 Ma has a below-average number of observations per time bin, suggesting it is relatively undersampled. *(b)* To improve the signal reconstruction, additional new samples (yellow) are collected from the undersampled interval in *a*, which is mapped back to each stratigraphic section using the posterior age models. The original samples (red) were collected at regular 1 m stratigraphic sampling intervals.

A straightforward strategy for improving the proxy signal inference is collecting additional proxy data. The model outputs can be used to identify time intervals where the proxy signal may be poorly constrained because observations are sparse or absent. These 'gaps' in the proxy data are prime targets for future data acquisition efforts. The *temporal density* of proxy
observations, defined as the number of observations within different user-defined time bins (averaged across all posterior draws), serves as a guide for identifying undersampled time intervals (Fig. 14a). Intervals that contain an average of one or fewer data points are particularly poorly constrained, and future data acquisition efforts should aim to fill these gaps. Intervals that contain a below-average number of data points also represent useful targets for future sampling, depending on the severity of the disparity.
With respect to the stratigraphic observations, temporal gaps in the data represent either hiatuses or undersampled parts of the stratigraphy. Undersampling often occurs when samples are collected at regular stratigraphic intervals (e.g., every 1 m) in

the field, but the mapping of time to stratigraphic height is irregular. As a result, intervals of time that are stratigraphically condensed (i.e., intervals with low sedimentation rate) may be sampled at a low temporal resolution or missed entirely. In cases where a gap in the data is caused by undersampling, the gap may be filled via targeted higher-resolution sampling of existing stratigraphic sections. Gaps that are caused by hiatuses can only be amended by collecting data from new locations with different sediment accumulation histories.

A workflow for identifying temporal gaps in the data, mapping these gaps to existing stratigraphic sections using their posterior age models, and conducting targeted sampling to fill these gaps is illustrated in Fig. 14. In this example, three synthetic sections record carbonate $\delta^{13}$C from 215 to 200 Ma. Each section experiences relatively slow sediment accumulation from $\sim$212 to 208 Ma, coincident with a $+7‰$ $\delta^{13}$C excursion. Because samples are collected at regular 1 m intervals in the field, the condensed excursion interval is only sampled twice in section 1, once in section 2, and is missed entirely in section 3 (Fig. 14b). Without prior knowledge of the age model for each section, we can detect this gap in the data by observing that the density of proxy observations from 215 to 207.5 Ma is relatively low, with two 0.5 Myr intervals averaging less than one observation (Fig. 14a). By mapping this undersampled time interval back to each section, we identify target stratigraphic intervals for higher-resolution sampling (Fig. 14b).

### 4.3.2 Measuring multiple proxies

In addition to collecting additional data for existing proxies, the inference can be improved by considering data for new proxy systems. In the inference model, the proxy observations provide information about the age and alignment of stratigraphic sections. Incorporating data for multiple proxies that may record global signals offers additional constraints on the section age models, and should lead to more accurate proxy signal reconstructions. In Sect. 3.2.2, for example, each additional proxy results in more accurate estimates of excursion timings and magnitudes, narrower 66% and 33% probability envelopes for the inferred proxy signals, and better synthetic proxy signal recovery (Fig. 9c-d). In some cases, multiproxy inference also can be leveraged to screen for diagenesis (see discussion in Sect. 4.1.2).

To use additional proxy systems to improve the accuracy of the inference, multiple proxies must have some stratigraphic overlap in at least one stratigraphic section. These overlapping segments allow information from other sections, which may or may not include observations for all of the proxies, to be included in the analysis. In the most basic sense, this addition of data provides more information about the real world to the model, and should lead to more accurate results. For example, imagine that a stratigrapher is reconstructing carbonate $\delta^{13}$C over time using observations from six stratigraphic sections (as in Sect. 3.2). Three of these sections are from mixed carbonate-siliciclastic basins, and also have data for the carbon isotopic composition of organic matter, $\delta^{13}$C$_{org}$. The stratigrapher adds these $\delta^{13}$C$_{org}$ data to the model, along with two new $\delta^{13}$C$_{org}$ profiles from purely siliciclastic basins. The three sections with both carbonate $\delta^{13}$C and $\delta^{13}$C$_{org}$ observations provide a critical link between the two proxy records: through the correlation process, the age models for *all* sections – not just those with observations for both proxies – will be informed by the new $\delta^{13}$C$_{org}$ data. This link is particularly valuable when it allows us to incorporate geochronological age constraints that otherwise could not be considered. For instance, a shale Re-Os age from one of the siliciclastic basins now could help to calibrate the reconstruction of carbonate $\delta^{13}$C over time.

### 4.3.3 Removing noisy sections and diagenetically altered samples

Sections with high inferred geologic noise terms ($\eta_{\text{section}}$) likely have been significantly influenced by non-global processes. These non-global processes can be fingerprinted by interrogating the model posterior in the context of other available geological and geochemical observations (Sect. 4.1.2). In general, the accuracy of the inferred common signal should not be degraded by 'noisy' sections, because the model has inferred that these sections are decoupled from the common proxy signal. However, if all or most sections have high geologic noise, then the model may be unable to recover information about past biogeochemical cycling because the primary signal has been entirely obscured. In addition, models that include many noisy sections (e.g., high-noise sections outnumber low-noise sections) may lead to proxy signal inferences that are positively or negatively biased. For example, if diagenesis has lowered proxy values in all of the noisy sections, then these negatively biased data may 'drag' the inferred proxy signal toward lower values in order to minimize the values of $\eta_{\text{section}}$ and $\phi$ required to explain the observations (Eq. 3). While inferred relative changes in proxy values over time (i.e., the shape of the proxy signal) should be unaffected, the inferred signal may be shifted up or down. In cases where the absolute proxy values are important, we recommend re-running the inference without very high-noise sections as a precaution.

Even if noisy sections do not degrade or bias the proxy signal reconstruction, sections with an average posterior $\eta_{\text{section}}$ term that is equal to or greater than the amplitude of the inferred proxy signal do not help to constrain the common proxy signal (as in the synthetic experiments where high-amplitude white noise was added to the proxy data; Sect. 3.3.1). Therefore, high-noise sections can be excluded from future analyses without losing any information. This removal will shorten sampling time for future model runs, and/or liberate computational resources so that potentially more informative sections can be included in the inference (in cases where a large data set has been downscaled to make the inference tractable; Sect. 2.8).

In cases where diagenesis is responsible for high inferred geologic noise (Sect. 4.1.2), it is possible that only a subset of samples have been severely altered, while the remaining samples encode valuable information about proxy change over time. However, the model may be unable to use this information because any coherent stratigraphic trends have been masked by the altered samples. For data-limited problems, we therefore suggest screening samples from noisy sections for diagenesis using independent geochemical, petrographic, and textural criteria. For carbonates, coupled changes in different stable isotope values ($\delta^{13}$C, $\delta^{44/40}$Ca, $\delta^{53}$Cr, $\delta^{7}$Li, $\delta^{26}$Mg, $\delta^{18}$O, $\delta^{34}$S, $\delta^{238}$U) and trace element concentrations (I, Mg, Mn, Sr, U) can be used to evaluate whether preserved proxy values reflect the chemistry of primary seawater or secondary fluids (e.g., Fantle and Higgins, 2014; Ahm et al., 2018; Fantle et al., 2020; Lau and Hardisty, 2022; Murphy et al., 2022). Within our modeling framework, correlated changes in these diagenetic indicators and the residuals between observed proxy values and the inferred proxy signal can be used to recognize diagenesis (as discussed in Sect. 4.1.2). Samples that are found to be severely altered should be excluded from the inference, while relatively well-preserved samples can be retained.

### 4.3.4 Considering sedimentological observations

Information about paleoenvironment – typically derived from detailed sedimentological observations – sometimes can be leveraged to improve proxy signal reconstructions. For example, in Sect. 3.2.3 we consider synthetic $\delta^{13}$C profiles from a

carbonate platform where different depositional environments impart distinct biases on proxy values. Before documenting the sedimentology of each section, we might naively assume that each location represents a different environment, and choose to use per-section offset terms in the model (recall that the offset term, $\phi$, captures local shifts relative the common signal). Because the depositional environment within each section – and thus its local bias – actually is not constant over time, the model is unable to capture the true offset between each sample and the common signal. Consequently, the inferred common signal may be biased. In our experiment, for example, $\delta^{13}$C is broadly overestimated and the $\delta^{13}$C signal inference has wide probability envelopes (Fig. 11d). When we instead use the sedimentology to group samples by depositional environment, the signal inference is more accurate and less uncertain (Fig. 11b) because our offset parameterization is consistent with the natural system, enabling the model to infer the correct offset for each environment (Fig. 11c).

We recommend grouping samples within a given basin by depositional environment wherever the requisite stratigraphic observations are available. Because the model learns from the data, this added context will not bias the results; if there are no environment-dependent offsets in the data, then the inferred offset for each environment will be zero. In this case, one might consider alternative (e.g., per-section or per-basin) offset groupings, based on prior knowledge or hypotheses about the local processes that may influence a given proxy. In general, the model that best approximates the natural system will produce 1) non-zero inferred offsets with comparatively low posterior variance, and 2) proxy signal inferences with comparatively low variance (i.e., narrower probability envelopes). If different candidate models yield similar results, then the choice of offset parameterization likely is unimportant.

While environmental offset groupings can lead to more accurate signal reconstructions (if environmental biases exist in the data), they do not guarantee that the inferred proxy signal will be entirely free of environmental biases. Importantly, biases in the geographic or environmental composition of the entire data set can degrade the accuracy of the proxy signal reconstruction. For example, if the environmental distribution of the proxy observations changes over time (e.g., all samples older than 200 Ma come from shallow-water carbonate platforms, while all samples younger than 200 Ma come from deep ocean basins), then the model will be unable to distinguish between environmental biases and temporal changes in global proxy values. To minimize environmental bias in the reconstructed proxy signal, the inference should include observations from many different basins and paleoenvironments whenever possible.

Detailed stratigraphic work also may lead to the recognition of correlative features that directly constrain the alignment among sections. For example, distinct stratigraphic patterns and surfaces that can be traced between or independently recognized in different sections may be used to construct a sequence stratigraphic framework for a basin. Within this framework, chronostratigraphic markers that are present in multiple stratigraphic sections, such as sequence and parasequence boundaries, provide relative age constraints. Similarly, confidently identified marker beds (e.g., diamictites associated with regional or global glaciation; Hoffman and Li, 2009) may inform the alignment of sections within and/or between basins. Both dated and undated correlative features can be encoded as age constraints in the model, as described in Sect. 2.4.

### 4.3.5 Number and depositional environment of sections

In Sect. 3.3.2, we demonstrate that the quality of proxy signal reconstructions depends on both the completeness and number of stratigraphic sections included in the model. For a fixed number of sections, signal recovery generally improves as deposition becomes more constant (i.e., high-$k$ sections in Fig. 12b). This result matches the conventional wisdom that data from low-energy environments where sedimentation is relatively continuous, such as deep open-marine basins, are the most informative. As such, we recommend that data from such environments (e.g., deep-sea sediment cores) should be considered wherever possible. However, deep-sea records are more rare in the geologic record than marginal sediments, and are more costly to obtain if drilling is required. As such, in practice it often is necessary to consider data from environments with more complex depositional histories.

While deep-sea records have simpler age-height relationships than shallow-water sediments, our experiments suggest that shallow-water records still preserve much of the same information. The model developed in this paper can be used to see through the more complex depositional histories of shallow-water sections in order to access this information. In our synthetic experiments, the quality of signal reconstructions that incorporate many sections with episodic depositional histories (low-$k$) rivals that of reconstructions that include fewer continuous (high-$k$) sections (Fig. 13b). In all cases, reconstruction accuracy increases with both the number of sections considered and the continuity of the depositional histories (Fig. 13c). These results reflect the inherent limitations of reconstructing signals from the sedimentary record: only those features (e.g., proxy excursions) that are preserved in at least one section can be recovered, and only sections with some temporal overlap can be correlated. When individual stratigraphic sections are relatively incomplete (low-$k$), considering a larger number of sections with unique depositional histories increases the likelihood that both of these conditions will be met. Therefore, in cases where observations are limited to shallow-water environments we suggest maximizing the number and geographic diversity of stratigraphic sections.

## 4.4 Modeling challenges

### 4.4.1 Sensitivity to model priors

Most model parameter priors are weakly informative, meaning that their influence on the posterior is minimal. However, the proxy signal inference is sensitive to two user-specified model components that should be selected carefully based on prior knowledge about the geology and the processes influencing the proxy of interest: the choice of which samples share an offset term ($\phi$), which is covered in Sect. 4.3.4, and the prior distribution for the Gaussian process covariance kernel lengthscale, which we discuss in this section.

The lengthscale of covariance controls the smoothness of the inferred proxy signal, where larger lengthscales correspond to lower-frequency, smoother signals and higher lengthscales correspond to more 'wiggly', high-frequency signals. When choosing a lengthscale prior for a particular problem, the expected timescale of variation for the proxy of interest should be considered. For example, resolving changes in seawater $\delta^{18}O$ during Pleistocene glacial cycles requires considerably shorter

lengthscales (corresponding to $<100$ kyr timescales) than resolving secular changes in seawater $\delta^{44/40}$Ca, which occur on timescales longer than the $\sim 1$ Myr oceanic residence time of calcium.

We recommend specifying the lengthscale prior such that very low lengthscales are not allowed (the appropriate lower bound depends on the proxy and timescale of interest, as discussed above). If the lengthscale is shorter than the spacing between proxy observations (i.e., the signal contains multiple 'wiggles' between adjacent data points), then two problems can arise (as described by the Stan Development Team, 2024). First, the model will overfit the data because the GP perfectly captures all of the proxy observations with zero variance. As a result, unrealistic high-frequency solutions – possibly resembling random noise

– will have inflated posterior probabilities. Second, different high-frequency solutions will have equal likelihoods because all of them can perfectly explain the proxy observations. This 'likelihood plateau' is problematic for MCMC sampling algorithms that use likelihood gradients to explore the posterior, including the No-U-Turn Sampler (Hoffman and Gelman, 2014). Conversely, if only very long lengthscales are allowed (relative to the timescale of interest), then the signal will be forcibly flattened, obscuring and blurring proxy changes. This flattening also produces a likelihood plateau because all long-lengthscale solutions

have equally poor explanatory power. For a given problem, the lengthscale prior must be carefully constrained such that it captures the full range of geologically reasonable solutions while avoiding these pitfalls. In practice, we find that a Wald distribution (as in Fig. 4a) often is a good choice of prior because it is positively skewed, which means that its hyperparameters can be specified such that the mode is centered around some reasonable intermediate lengthscale, the probability of very short lengthscales approaches zero, and longer lengthscales are still allowed. To explicitly force a specific minimum lengthscale, the

prior distribution can be manually translated by a fixed value.

### 4.4.2    Scalability of the inference model

To a first order, the computational complexity of the model is controlled by the $\mathcal{O}(n^3)$ scaling of exact Gaussian process inference, where $n$ is the number of proxy observations (Rasmussen et al., 2006). For a given model, sampling time scales linearly with the number of steps taken during each Markov chain simulation (Sect. 2.7.1). Due to the difficulty of explor-

ing potentially multimodal posterior distributions (Sect. 2.7.2), we find that it is generally more advantageous to run many independent Markov chains than to run individual simulations for longer; the recommended default is $2,000$ steps (with the first $1,000$ discarded for sampler burn-in). Different Markov chains can be run in parallel, where one core is allocated to each chain. Depending on computational resources, the inference may prove intractable for more than several hundred observations; in Appendix B, we describe a Gaussian process approximation method that reduces the computational complexity such that it

scales linearly with $n$.

### 4.5    Limitations of chemostratigraphy

The modeling framework developed in this paper offers an objective and reproducible way to reconstruct past changes in regional or global biogeochemical cycling from stratigraphic observations. However, the accuracy of the inference hinges on the validity of the assumptions encoded in the model prior and on the quality of the observations (see, for example, our

discussion of biased data sets in Sect. 4.3.4). Importantly, the model itself does not elucidate the nature of the inferred proxy

signal; it only isolates the trends in proxy values that are common to all stratigraphic sections. While this common signal may represent large-scale biogeochemical cycling, it also could reflect a number of unrelated processes that are capable of imparting similarly coherent stratigraphic trends in proxy values. In this section, we provide a few examples of such alternative processes and consider how we might test different hypotheses for the processes driving proxy change.

Observations from the rock record offer a warning that, in some cases, the common proxy signal recovered by the inference model might represent diagenetic or primary processes that occur simultaneously in all or many locations, but that are unrelated to large-scale biogeochemical cycling. For instance, Pleistocene carbonates from deep-water environments adjacent to carbonate platforms commonly record high-frequency, $3 - 4‰$ $\delta^{13}C$ excursions that reflect globally synchronous changes in shallow-water sediment production and transport during glacial cycles while global-mean seawater $\delta^{13}C_{DIC}$ remains nearly

constant (Swart, 2008; Edmonsond et al., 2024). In adjacent shallow-water environments, glacioeustatic sea level fall facilitates widespread meteoric alteration of exposed carbonate platforms, creating similar stratigraphic $\delta^{13}C$ profiles in many locations (Allan and Matthews, 1982; Melim et al., 2001; Swart and Eberli, 2005). Analogous episodes of widespread platform exposure and alteration have been inferred during the late Paleozoic ice age (Bishop et al., 2009; Dyer et al., 2015). More controversially, some workers have proposed a meteoric or burial diagenesis origin for the Neoproterozoic Shuram negative

carbon isotope anomaly (Knauth and Kennedy, 2009; Derry, 2010). In each of these examples, the reconstructed proxy signal does not reflect primary seaweater chemistry, but it does still encode information about synchronous, large-scale Earth system change. However, it is important to note that if age control is poor, then non-synchronous local processes that produce similar proxy profiles – for example, different episodes of meteoric alteration related to local uplift – potentially can lead to spurious correlations and incorrect proxy signal reconstructions (Smith and Swart, 2022).

Determining whether a reconstructed signal represents large-scale biogeochemical cycling often requires measuring multiple geochemical proxies (e.g., carbonate $\delta^{13}C$, $\delta^{18}O$, $\delta^{44/40}Ca$, and trace element concentrations) that can be used to fingerprint different processes. While recognition of diagenesis is aided by a host of quantitative models (Fantle and Higgins, 2014; Ahm et al., 2018; Fantle et al., 2020; Lau and Hardisty, 2022; Murphy et al., 2022), determining whether a proxy signal represents large-scale biogeochemical cycling or unrelated syndepositional conditions remains challenging. In some cases, linking

sedimentological observations to the model outputs may help to distinguish between different hypotheses. For example, the distribution of depositional environments over time may give insight to whether an excursion could be driven by synchronous stratigraphic changes in depositional environment (e.g., during sea level change) rather than secular changes in seawater chemistry.

## 4.6 Implications for reconstructing past Earth system change

The specific magnitude, duration, and rate of geochemical proxy change can directly test various hypotheses about past biogeochemical cycling. For example, estimates of durations and rates can aid in evaluating different hypotheses for the cause of events such as mass extinctions (Schobben et al., 2019; Song et al., 2021), episodes of global warming or cooling (Zachos et al., 2010; Finnegan et al., 2011), and oxygenation of the oceans and atmosphere (Kah et al., 2004; Algeo et al., 2015). However, durations and rates sometimes are reported without uncertainties, which hampers hypothesis testing when investigating

ancient events with limited absolute age control. In addition, disparate approaches to age model construction (e.g., classical versus Bayesian methods) can hinder or introduce bias to comparisons between different events. For instance, Reershemius and Planavsky (2021) compare the durations of Mesozoic and Paleozoic ocean anoxic events (OAEs) using previously published age models that are largely derived from cyclostratigraphy during the Mesozoic, and from a host of different approaches (e.g., linear interpolation between radiometric ages, chemostratigraphy, sedimentation rate estimates) during the Paleozoic. As a re-

sult, it is difficult to distinguish between real differences in rates of Earth system change and potentially specious differences stemming from incongruent age modeling approaches. The model developed in this paper is the first tool that can objectively and reproducibly estimate the magnitude, duration, and rate of past proxy perturbations. Furthermore, multiproxy correlation can be leveraged to place different proxy records (e.g., $\delta^{13}$C, $\delta^{18}$O, and $\delta^{34}$S) in the same temporal framework, which alleviates issues that arise when comparing proxy records that were constructed separately.

Our model also provides a new framework for detecting and reconstructing spatial gradients in past seawater chemistry and deconvolving these patterns from global proxy excursions (Sect. 4.1.1). For example, it has been proposed that Cambrian (Schiffbauer et al., 2017) and Ordovician (Yang et al., 2024) $\delta^{13}$C excursions may be modulated or caused by water depth $\delta^{13}$C$_{\mathrm{DIC}}$ stratification. Explicitly deconvolving the spatial and temporal components of these excursions can improve reconstructions of past $pO_2$ and $pCO_2$ levels that are calibrated using marine $\delta^{13}$C records (Berner, 2006; Saltzman and Edwards,

2017; Krause et al., 2018). Similarly, reconstructing gradients in paleotemperature proxies such as $\delta^{18}$O, Mg/Ca, and $\Delta_{47}$ could improve reconstructions of past ice volume, vertical temperature profiles, and sea surface temperature patterns (Finnegan et al., 2011; Jones and Eichenseer, 2021; Grossman and Joachimski, 2022).

Finally, our model is particularly well-suited for reconstructing proxy change during intervals of Earth history where individual records are incomplete, variably altered, and potentially locally biased. For example, the timing, magnitude, and structure

of the Paleoproterozoic Lomagundi-Jatuli carbon isotope excursion remain poorly understood (Hodgskiss et al., 2023), due in large part to the difficulty of merging incomplete, sparsely dated, and heterogeneous $\delta^{13}$C records from different locations. A quantitative reconstruction of $\delta^{13}$C over time (with uncertainty) could help to distinguish between competing hypotheses for the excursion's cause, global versus local nature, and relationship with the rise of atmospheric oxygen.

## 5 Conclusions

In this paper, we developed a Bayesian inference model for reconstructing past changes in global biogeochemical cycles that are recorded as geochemical proxies in ancient sedimentary rocks. This model improves on previous approaches by explicitly untangling global and local signals, coupling chemostratigraphic correlation with age model construction, tracking uncertainty in all model parameters, and simultaneously inferring global changes in multiple proxies. Synthetic case studies confirm that our model can accurately reconstruct proxy change over time even when age constraints are sparse, proxy records have been

biased by local processes or overprinted by diagenesis, and the relationship between stratigraphic height and time is highly irregular. However, the real explanatory power of the model comes from situating the inference results in their geologic and paleoenvironmental context as part of the scientific process. Future applications of the model to observations from ancient

shallow-water environments will yield highly testable reconstructions of past Earth system change that more accurately capture the intrinsic uncertainties associated with reading a fragmented and noisy sedimentary record.

*Code and data availability.* Code for the version of StratMC developed in this paper is archived on Zenodo (https://doi.org/10.5281/zenodo.13324359; Edmonsond, 2024a), along with a supplementary *User Manual*. The model outputs for all synthetic experiments are archived in a separate Zenodo repository (https://doi.org/10.5281/zenodo.13119724; Edmonsond, 2024c), along with the code required to reproduce our results. The current version of the model is available on both GitHub (https://github.com/sedmonsond/stratmc) and Zenodo (https://doi.org/10.5281/zenodo.13281935; Edmonsond, 2024b), and associated package documentation is available at https://stratmc.readthedocs.io/.

## 805 Appendix A: Evaluating posterior stability

In Sect. 2.7.2 of the main text, we describe how Markov chain Monte Carlo (MCMC) samplers can struggle to converge when sampling complex and multimodal posterior distributions. Here, we provide a practical workflow for 1) recognizing inferences with multimodal posterior distributions, and 2) assessing whether multimodal posterior distributions have been thoroughly explored during sampling (i.e., the inference has stabilized).

In all cases, we recommend sampling the model posterior with at least eight independent Markov chains to ensure that posterior multimodality, if present, can be detected. This recommendation is based on empirical tests that suggest inferences with fewer than eight chains are more likely to be unreliable; however, if additional computational resources are available, then running more chains can only improve exploration. For a given inference, multimodal posterior distributions then can be identified by visually inspecting trace plots – which show the evolution of parameter values during each Markov chain 815 simulation – for key model parameters. In practice, the posterior distribution for the RBF kernel lengthscale hyperparameter often is particularly informative because different lengthscale modes correspond to proxy signals with different frequencies (Sect. 4.4.1). In Fig. A1a, trace plots for the lengthscale hyperparameter reveal that while each Markov chain is stationary, or stable, the different chains have not mixed, meaning that each chain has explored a different part of the posterior. Poor mixing indicates that the posterior distributions are multimodal, and each chain is 'stuck' in a different mode. Note that in cases where 820 individual chains 'jump' between modes during sampling, the chains may also not be stationary. In contrast, the chains in Fig. A1b are both stationary and mixed, indicating that they have converged to the same unimodal posterior distribution.

For models with multimodal posterior distributions, we recommend running at least 20 Markov chains in parallel to ensure that the posterior parameter space is thoroughly explored. To check that the inference is stable – meaning that considering additional chains does not affect the results – key 'stability metrics' then should be evaluated. Recommended metrics can 825 be calculated and visualized using functions provided in the `inference` and `plotting` submodules; refer to the package API in the *User Manual* and the online documentation. The first metric is the standard deviation of the inferred RBF kernel lengthscale hyperparameter, which stabilizes once all significant posterior modes have been sampled. In Fig. A1c, for example, the standard deviation increases rapidly between 0 and 10 chains – indicating that each additional chain is exploring a new part of the parameter space – and stabilizes after $15 - 20$ chains have been considered, which indicates that new modes are

## Identifying multimodal posterior distributions:

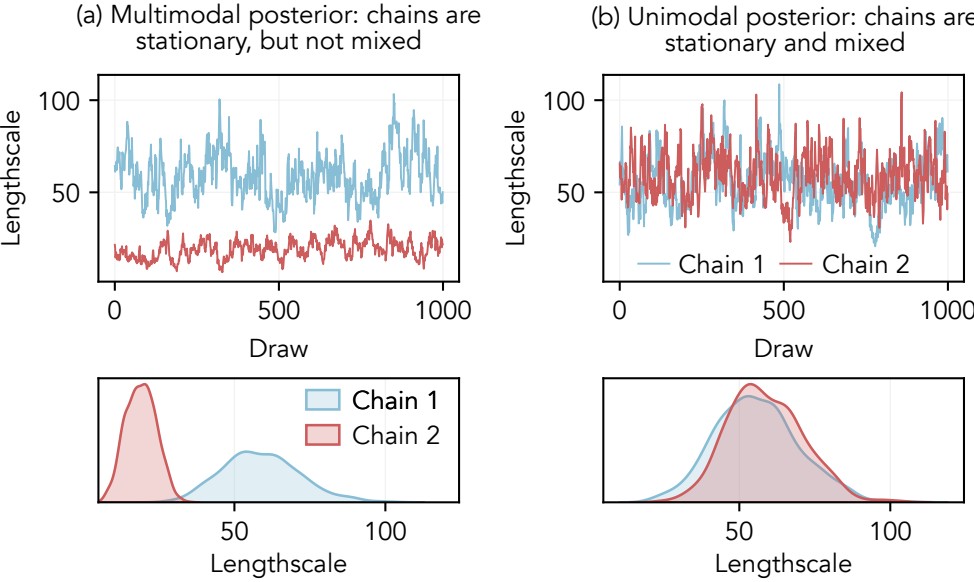

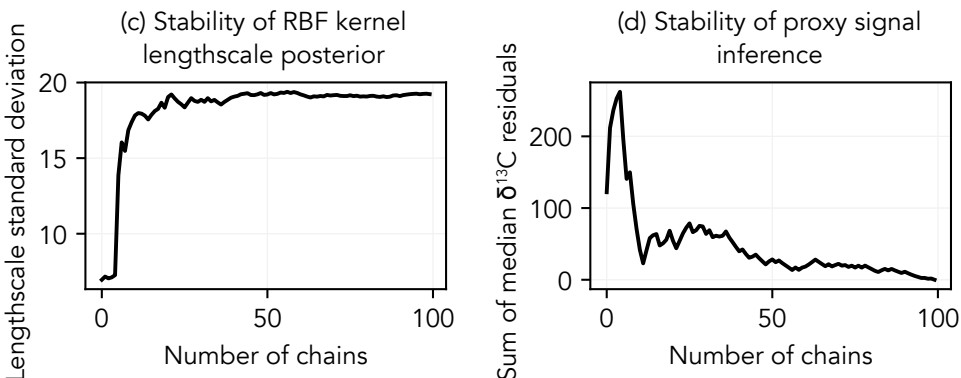

## Evaluating stability for models with multimodal posterior distributions:

**Figure A1.** Recognizing posterior multimodality and assessing the stability of the inference for models with multimodal posterior distributions. *(a, b)* Posterior draws for the covariance kernel lengthscale hyperparameter. In *(a)*, each chain has explored a different mode of the posterior distribution (each chain is stationary, but the chains have not mixed); in *(b)*, both chains have converged to the same posterior distribution (the chains are both stationary and mixed). *(c)* Posterior standard deviation of the covariance kernel lengthscale hyperparameter when 1 to 100 chains are considered; each chain contains $1,000$ draws. *(d)* Sum of residuals between the median posterior proxy value (evaluated at each age) for 100 chains versus 1 to 99 chains.

no longer being discovered. The second metric is the stability of the proxy signal inference. To evaluate whether the proxy signal posterior has been adequately explored, we calculate the sum of residuals between the median posterior proxy value (evaluated at each age) for all $N$ chains versus 1 to $N-1$ chains. Put simply, this metric measures 'how different' the proxy signal inference is when a subset of $n$ chains is considered versus when all $N$ chains are considered. If considering additional chains does not significantly change the proxy signal inference, then the residuals will approach zero and stabilize. In Fig. A1d,

the inference has largely stabilized after approximately 20 chains have been considered, and more definitively stabilized after 50 chains have been considered. For real problems, additional chains should be run if the inference has not stabilized after the initial chains have been considered.

## Appendix B: Working with large data sets: Gaussian process approximations

In Sect. 2.8 and 4.4.2 of the main text, we state that inference can become intractable for data sets that include more than several

hundred proxy observations due to the computational complexity of exact Gaussian process (GP) regression. A number of GP approximations have been developed to make GP regression tractable for large data sets (Rasmussen et al., 2006). One of these methods is the reduced-rank Hilbert space Gaussian process (HSGP) approximation, which approximates the GP as a linear model defined by $m$ basis functions (Solin and Särkkä, 2020). The computational complexity of the HSGP approximation scales as $\mathcal{O}(mn + m)$ (Riutort-Mayol et al., 2022); as a result, the HSGP approximation can handle larger data sets than an

unapproximated GP, which scales poorly with computational complexity $\mathcal{O}(n^3)$. For an extended theoretical explanation of the HSGP approximation, see Solin and Särkkä (2020).

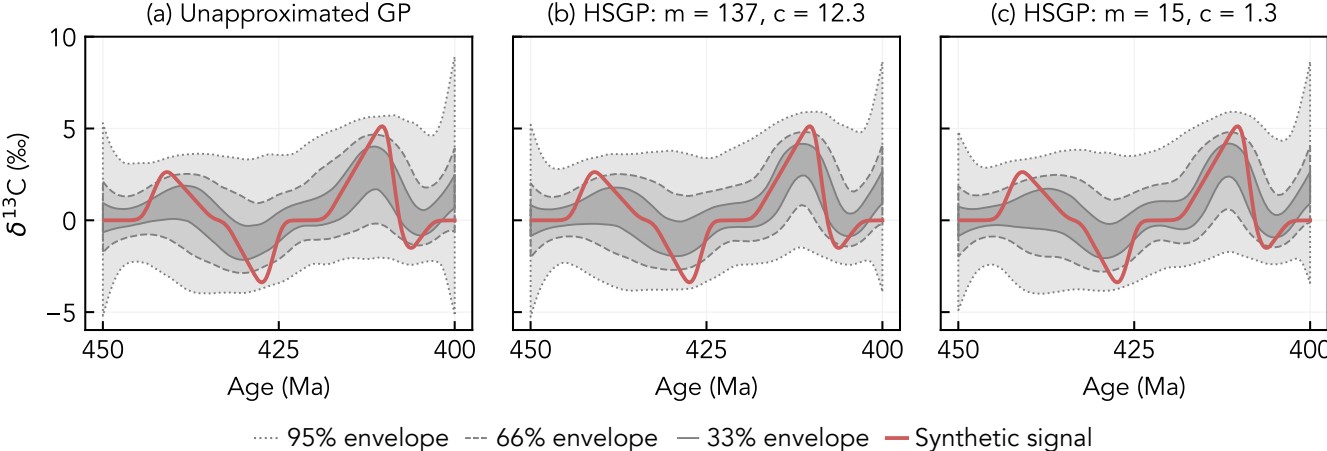

**Figure B1.** Inferred $\delta^{13}$C signals using *(a)* the unapproximated Gaussian process model, *(b)* the Hilbert space approximation with $m = 137$ and $c = 12.3$ (the recommended parameters based on the range of the RBF kernel lengthscale prior), and *(c)* the Hilbert space approximation with $m = 15$ and $c = 1.3$ (the recommended parameters based on the range of the RBF kernel lengthscale posterior from the unapproximated inference in *a*). Stratigraphic proxy observations and age constraints are as in Fig. 9b.

We find that proxy signal inferences that use the HSGP approximation are indistinguishable from those that use an unapproximated GP (Fig. B1). However, the accuracy of the approximation depends on two user-specified parameters: $m$, the number of basis functions used to approximate the proxy signal, and $c$, which controls the interval over which the approximation is valid (Solin and Särkkä, 2020). Functions with shorter lengthscales (i.e., more 'wiggly' or high-frequency proxy signals) require more basis functions (high $m$), while functions with longer lengthscales can be accurately approximated with fewer basis functions (lower $m$). In practice, $m$ and $c$ must increase in tandem to maintain the fidelity of the approximation (Riutort-Mayol et al., 2022). Due to the $\mathcal{O}(mn + m)$ scaling of the HSGP approximation, the reduction in computational complexity (compared to an unapproximated GP) will be more significant for lower $m$. As a result, if $m$ is sufficiently large then the HSGP approximation may be less efficient than the unapproximated GP for small data sets.

The `pymc.gp.hsgp_approx` module in PyMC (version 5.16.2; Abril-Pla et al., 2023) has the method `approx_hsgp_hyperparams`, which uses formulas developed in Riutort-Mayol et al. (2022) to estimate optimal values for the $m$ and $c$ parameters given the total timespan of the proxy signal and the range of possible RBF kernel lengthscales. In the package documentation (https://stratmc.readthedocs.io/), we provide example code for using the HSGP approximation, including tuning the $m$ and $c$ parameters. Note that high values of $m$ and $c$ are required to ensure the approximation is valid for a wide range of possible lengthscales. As a result, sampling efficiency may be improved by more tightly constraining the range of probable RBF kernel lengthscales (by e.g., running an unapproximated inference using a downsampled version of the data). For example, in Fig. B1, sampling the HSGP model with $m = 137$ and $c = 12.3$ (recommended parameters based on the full range of the lengthscale prior) is 1.7 times faster than sampling the unapproximated GP model, while the model with $m = 15$ and $c = 1.3$ (recommended parameters based on the posterior lengthscale range from the unapproximated inference) produces identical results and samples 10.5 times faster than the unapproximated GP model. For more comprehensive guidance on tuning the HSGP parameters, refer to Riutort-Mayol et al. (2022).

*Author contributions.* BD and SE conceptualized and designed the model. SE wrote the model code, designed and executed the experiments, and drafted the manuscript with input and supervision from BD.

*Competing interests.* The authors declare that they have no competing interests.

*Acknowledgements.* We thank Anne-Sofie Ahm, Roberta Hamme, and Terri Lacourse for insighftful feedback and discussion on an early version of this manuscript. This research was funded by a Natural Sciences and Engineering Research Council of Canada (NSERC) Discovery Grant to B. Dyer (RGPIN-2021–04082).

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
