# Peer review of "A Bayesian framework for inferring regional and global change from stratigraphic proxy records (StratMC v1.0)"

_EGUsphere, 2024_

## Referee Comment (RC1)

A Bayesian framework for inferring regional and global change from stratigraphic proxy records (StratMC v1.0)

criteria

| Principal criteria | Excellent (1) | Good (2) | Fair (3) | Poor (4) |
|---|---|---|---|---|
| **Scientific significance:** Does the manuscript represent a substantial contribution to modelling science within the scope of Geoscientific Model Development (substantial new concepts, ideas, or methods)? | | 2 | | |
| **Scientific quality:** Are the scientific approach and applied methods valid? Are the results discussed in an appropriate and balanced way (consideration of related work, including appropriate references)? Do the models, technical advances, and/or experiments described have the potential to perform calculations leading to significant scientific results? | | 2 | | |
| **Scientific reproducibility:** To what extent is the modelling science reproducible? Is the description sufficiently complete and precise to allow reproduction of the science by fellow scientists (traceability of results)? | 1 | | | |
| **Presentation quality:** Are the methods, results, and conclusions presented in a clear, concise, and well-structured way (number and quality of figures/tables, appropriate use of English language)? | 1 | | | |

**General Comments**

The authors provide a novel Bayesian approach for simultaneous absolute age modeling and correlation of stratigraphic sections with multiple proxies. This approach is based on the inference of a "common signal" shared by proxies (on a proxy-by-proxy level), which may be biased in individual sections. The model is clearly presented, and the code implementing the model is well-documented and easy enough to run. The authors validate the model with

synthetic examples primarily focusing on d13C in carbonates, for which they explore various scenarios of signal recovery via their model. While I find that the correlation aspect of their model is appropriate for the applications they envision, my main feedback is that the age modeling (upon which correlation is dependent, as they are simultaneously modeled) is ad-hoc and requires a stronger basis in the statistics of the distribution of time in stratigraphy. Similarly, the experiments exploring "temporal noise" require firmer grounding in theory. See subsequent specific comments for more details. I believe that this concern can be addressed with deeper interrogation of the age modeling priors and/or reformulation of the priors to more deliberately incorporate our knowledge of the temporal statistics of stratigraphy. The authors also need to better build the intuition for how the age model priors in conjunction with age constraints affect the output of the Bayesian model. My other comments address more minor issues, including more thorough comparison with the Bayesian model of Lee et al. (2023) and the utilization of reliability diagrams as an additional, more nuanced evaluation of their probabilistic model performance. Overall, with some revisions along the lines elaborated on below, I think this manuscript provides an important contribution as a modeling framework for correlation and age modeling, especially for the types of applications that the authors have highlighted.

**Specific Comments**

**Age modeling and correlation**

The authors clearly lay out various difficulties in both age modeling and stratigraphic correlation, and they also provide an intuition for how correlation can help with age modeling by constraining likely synchronous levels within various sections. The authors, however, fail to clearly establish *how* the information that goes into the age modeling propagates to the posterior inference for the common proxy, and how this information is tied to the fairly well-established statistics of the distribution of time in stratigraphy.

Imagine a common proxy signal that is simply a sine wave with a single period, which is perfectly recorded in several sections. This signal can be exactly correlated. Now imagine that the sections only contain precise age constraints at their tops and bottoms. In this case, even though the signals are trivially correlated, the location of the peak and trough of the sine in absolute time will be highly uncertain. With extremely ignorant priors on the age modeling, it's conceivable that the posterior distribution for the common proxy signal in absolute time may even have flat contours that completely encapsulate the amplitude of the sine. Ideally, the priors on the age modeling would ensure that the temporal structure of the posterior appropriately encapsulates the true structure. However, the figures that the authors present demonstrate that this is not the case for their model. For example, in Figure 8, the authors show that the posterior model nicely recovers the overall shape of true common proxy signal. However, the locations of peaks and troughs in absolute time seem (subjectively speaking) too tightly constrained, such that the posterior model significantly deviates from the true model at the locations of most major peaks and troughs. This result seems to be due entirely to the age modeling, for which the

priors appear to be *too* informative. This subjectively described behavior can be quantified with a reliability diagram (see a subsequent comment).

All this discussion brings me to the main point, which is that the authors need to more critically consider the prior age modeling. The authors state that they (line 138) "construct prior age models with the goal of imposing no limits on sedimentation rate between age constraints," and yet in Figure 3c they show the distribution resulting from their prior modeling approach. What is this distribution? Sadler (who the authors cite) demonstrated that the concept of sedimentation rate is only relevant at a timescale of interest, since the Sadler Effect shows that sedimentation rates decrease as a power law with averaging timescale. At a particular timescale of interest (within an order of magnitude or so), Sadler also demonstrated that sedimentation rates follow a log normal distribution. Figure 3c does not appear to be log normal. The ad-hoc approach that the authors have taken with the shift and scale parameters was probably motivated by modeling convenience, but it muddies the waters in terms of incorporating empirical statistical information about sedimentation rates. I recommend that the authors reformulate their priors for the age modeling. Specifically, the authors should explicitly consider how the sedimentation rates they model probabilistically are tied to a timescale (as they must be), which itself might be a random variable in their model. If the authors think that proxy sample spacing may span timescales over multiple orders of magnitude, they should grapple with the implications that makes for prior sedimentation rate modeling throughout a sampled section. I recommend abandoning the ad-hoc approach, which imposes a poorly interpretable prior. Finally, the prior that is ultimately chosen should result in age models that yield *reliable* posterior models (see subsequent point).

> As an aside, I could imagine that this modeling framework might be able to introduce an intermediate hidden variable that captures the well-constrained, correlated component of the common proxy signal, which exists along a coordinate that has a monotonic relationship with both absolute age *and* stratigraphic height in each section. This internal representation separates the correlation from the age modeling problem. The task would then be to evaluate the likelihood of the monotonic maps between age and height for each section, mediated via this hidden variable. These mappings would be informed by the prior assumptions about sedimentation rate as well as potentially any stratigraphic information indicating disconformity, etc.

**Quantifying model performance with reliability diagrams**

I appreciated the authors' mean signal likelihood metric, which captures the overall performance of their modeling approach in a single number. However, given that they are evaluating the performance of a probabilistic inference with respect to the truth, they should also utilize reliability diagrams, which show how the predicted distribution of values corresponds to the actually observed distribution. Bröcker and Smith (2007) (https://doi.org/10.1175/WAF993.1) provide a useful reference for constructing reliability diagrams with bootstrapped confidence intervals. I suspect the authors will find that the current

formulation of their model underestimates the true signal at both the lower and upper prediction quantiles due to the afore-mentioned over-confidence in the absolute time location of the common proxy signal. That is, I expect that the reliability diagrams for the current modeling approach would have low slopes falling off of the 1:1 line for a perfectly reliable model, in which case hopefully a modified age modeling prior would improve reliability.

In figure 9c, reliability diagrams may also reveal another slightly troubling result. The authors nicely show how the incorporation of multiple proxies significantly increases the synchronicity of the posterior with each true common proxy. However, the confidence intervals, especially at the low and high tails, do not seem to sufficiently collapse to reflect the improved modeling. Why might the model be overestimating uncertainty in the tails for inferences with more proxy systems?

**Comparison with Lee et al. (2023)**

While the authors do mention Lee at al. (2023) in the introduction, I think more can be done to compare the two modeling approaches. To my knowledge, Lee et al. (2023) provide the sole other Bayesian approach to simultaneous age modeling and (single) proxy correlation. Given that the authors are presenting exactly the same sort of model, they should be more explicit in acknowledging the similarities between the models and then highlighting the contributions they have made to this type of modeling, namely:

1. prior age modeling that is not strictly tied to assumptions about deep sea sedimentation rates (although as previously mentioned, this approach needs to be better grounded in theory) and
2. multi proxy correlation (which is mentioned in section 4.2, but with insufficient context)

Section 4.2 needs to reference Lee et al. (2023), and there could be a couple more sentences highlighting the similarities between the models either in the introduction or methodology. For instance, Lee et al. (2023) also permit inference of section-by-section offsets and variance scaling with respect to the inferred common proxy signal (albeit for a single proxy).

I think the authors should consider applying their model to the same d18O and radiocarbon dataset modeled by Lee et al. (2023) (perhaps just the Deep North Atlantic dataset, for example) as an application with real-world data and an opportunity for direct inter-model comparison. Several if not all of the cores utilized by Lee et al. (2023) in that stack have other proxy measurements (d13C, elemental concentrations, etc.) that could be utilized by the authors' new approach.

**"Non-uniform" depositional histories**

The treatment of "episodic" sedimentation could also be better grounded in the theory of time's distribution within stratigraphy. The Sadler Effect arises due to the power law distribution of hiatus within stratigraphy; the distribution of hiatus therefore dictates apparent sedimentation

rates. Hiatuses result from the dynamics (autogenics) of sedimentation as well as processes such as sea level, tectonics, etc. A critical timescale is the compensation timescale: below this timescale, stratigraphy is incomplete, which approximately corresponds to the "episodic" realm described by the authors. Beyond this timescale, stratigraphy is complete (the "continuous" regime), up until hiatuses resulting from longer timescale processes such as tectonic modifications of basin accommodation. What the authors refer to as "temporal noise" and "episodic sedimentation" are in fact the statistical structure of hiatus in stratigraphy, which results in stratigraphic incompleteness at short and long timescales.

This background brings me to the main point of this comment, which is that the authors need to take care that they are realistically modeling stratigraphy as best as we understand it when constructing their synthetic examples. For instance, by modeling the elapsed time between approximately evenly-spaced (in space) samples as a gamma distribution, do the resulting height increments ('the devil's staircase' shown in the right panel of Figure 6b) have a truncated/exponentially tempered power law distribution for small values of $k$, as we expect (Ganti et al. 2011)? How does the truncation of the power law (i.e., the compensation timescale) depend on the value of $k$? The authors should establish the theoretical connections between their current stratigraphic synthesis protocol and the relevant quantities in our current understanding of time's distribution in stratigraphy (such as the compensation timescale, the power law distribution of hiatus in stratigraphy, which truncates at intermediate (post compensational) timescales). Paola et al. (2019) provide a great review of these concepts (https://doi.org/10.1146/annurev-earth-082517-010129).

Alternatively, the authors could reformulate how they generate their synthetic stratigraphies. For example, rather than sampling time increments according to a gamma distribution, which appears to be an arbitrary decision unmotivated by theory, they could instead simulate stratigraphies with varying compensation timescales and stratigraphic completeness (i.e., hiatus power law exponents), which are then sampled regularly (or not) in space. The results of Section 3.3.2 would then be much easier to interpret with respect to the theoretical framework that exists for stratigraphy; perhaps the authors could modify Figure 13 to reflect various values of the hiatus power law exponent (completeness) and compensation timescale.

**Technical Comments**

- In Equation 4, the notation seems imprecise. Is it not in fact the evaluation of the posterior over $\theta$ at the true proxy value? $P_{\theta_{f(t_n)}}(g(t_n))$.
- Fig 1
  - would be nice to annotate the relevant parts with the notation introduced in Equation 1
  - empty box under model seems to serve no purpose
- fig 5
  - would be easier to compare panels a and b if the axes in panel a were flipped

- line 363: might be worth clarifying that white noise is independent, identically distributed, zero mean
- line 484-485: This sentence minimizes the importance of the age modeling procedure for the construction of the common proxy signal...it's not really like the age modeling is a byproduct of the authors' model. It's an integral part of the inference.
- line 567, 596: siliclastic -> siliciclastic

---

## Author Response (AR1)

**Response to Reviewer Comments*: A Bayesian framework for inferring regional and global change from stratigraphic proxy records (StratMC v1.0)**

Stacey Edmonsond[1] and Blake Dyer[1]

[1]School of Earth and Ocean Sciences, University of Victoria, Victoria, BC, Canada

**Correspondence:** Stacey Edmonsond (sedmonsond@uvic.ca)

We would like to thank Adrian Tasistro-Hart (R1) and one anonymous reviewer (R2) for their helpful and constructive comments on our manuscript. We have revised the manuscript to address the concerns raised by both reviewers. First, we incorporated extended clarification and discussion of our age modeling approach to address the concerns raised by R1. This discussion, coupled with our below response to R1's comments, demonstrates that our original age modeling approach is consistent with the recommendations of R1 and does not require reformulation. Second, we expanded our literature review to incorporate relevant references suggested by R2, and provided a more detailed comparison with the work of Lee et al. (2023) following the suggestion of R1. Finally, we addressed R1's concerns about our simulated depositional histories by clarifying how our methodology is consistent with existing stratigraphic modeling approaches.

Below, we provide a line-by-line response to the two reviews. The reviewers' comments are in blue, while our response is in black. Edits made to the manuscript to address the reviewers' comments are provided as **bold** block quotes. Line numbers are with respect to the revised manuscript (line numbers in the 'tracked changes' version are slightly offset).

**1 Response to RC1**

**1.1 General Comments**

The authors provide a novel Bayesian approach for simultaneous absolute age modeling and correlation of stratigraphic sections with multiple proxies. This approach is based on the inference of a "common signal" shared by proxies (on a proxy-by-proxy level), which may be biased in individual sections. The model is clearly presented, and the code implementing the model is well-documented and easy enough to run. The authors validate the model with synthetic examples primarily focusing on d13C in carbonates, for which they explore various scenarios of signal recovery via their model. While I find that the correlation aspect of their model is appropriate for the applications they envision, my main feedback is that the age modeling (upon which correlation is dependent, as they are simultaneously modeled) is ad-hoc and requires a stronger basis in the statistics of the distribution of time in stratigraphy. Similarly, the experiments exploring "temporal noise" require firmer grounding in theory. See subsequent specific comments for more details. I believe that this concern can be addressed with deeper interrogation of the age modeling priors and/or reformulation of the priors to more deliberately incorporate our knowledge of the temporal

statistics of stratigraphy. The authors also need to better build the intuition for how the age model priors in conjunction with age constraints affect the output of the Bayesian model. My other comments address more minor issues, including more thorough comparison with the Bayesian model of Lee et al. (2023) and the utilization of reliability diagrams as an additional, more nuanced evaluation of their probabilistic model performance. Overall, with some revisions along the lines elaborated on below, I think this manuscript provides an important contribution as a modeling framework for correlation and age modeling, especially for the types of applications that the authors have highlighted.

Thank you for the clear and thoughtful comments on our work; we particularly appreciate your careful consideration of our age modeling approach. We address your specific concerns in detail below. Importantly, we clarify that our age modeling is not ad-hoc, but rather based on our fundamental understanding of how sediment accumulates in nature. We further show that our age models are in fact consistent with existing knowledge about the distribution of time in stratigraphy. We recognize that because this fact was lost on careful read by R1, there was a failure in communication on our part. To fix that failure, we have revised and expanded the text to emphasize this point.

**1.2 Specific Comments**

**1.2.1 Age modeling and correlation**

The authors clearly lay out various difficulties in both age modeling and stratigraphic correlation, and they also provide an intuition for how correlation can help with age modeling by constraining likely synchronous levels within various sections. The authors, however, fail to clearly establish *how* the information that goes into the age modeling propagates to the posterior inference for the common proxy, and how this information is tied to the fairly well-established statistics of the distribution of time in stratigraphy.

We first address your overarching concern about clearly establishing how the information that goes into the age modeling propagates to the posterior inference for the common proxy signal. The manuscript addresses this question in two ways: first, the fundamental link between the prior and the posterior is established through our explanation of Bayes' theorem (Section 2.2). Then, Section 2.5 (Modeling sample ages) details how the prior age models are constructed, including the geologic assumptions embedded in our approach. Our revised manuscript includes more extensive discussion of these assumptions (see subsequent parts of our response). To highlight the connection between the prior and posterior age models, we also added the following sentence to Section 2.5 (directly after discussion of the assumptions encoded in the prior):

**The posterior age models are computed by merging these prior expectations with evidence in the data (Eq. 1).**

Beyond establishing this fundamental link between the prior and posterior, it is not possible to make general statements about exactly how the prior age modeling is ultimately propagated to the proxy signal inference. The degree to which the prior shapes the posterior fundamentally depends on how much information is gained from the observations (i.e., on the likelihood). If the data are uninformative – for example, if there is no common signal – then the section age models will not be further constrained

through correlation, and the posterior will match the prior. If the data are informative, then the posterior age models will have higher probability density within regions of the parameter space that facilitate optimal correlation among all sections (i.e., that maximize the likelihood within the bounds of the prior). We concur that the shape of the prior is an essential part of this calculation: that is the basis of any Bayesian model. However, given that the prior age models allow each sample to have *any* age that respects superposition with its bounding age constraints – meaning that the prior does not preclude any geologically possible solutions (although geologically unlikely scenarios have lower prior probabilities; see subsequent discussion) – how this information actually manifests in the proxy signal inference depends on the data. In other words, age models with low prior probability may have high posterior probability if the observations require that to be the case.

Imagine a common proxy signal that is simply a sine wave with a single period, which is perfectly recorded in several sections. This signal can be exactly correlated. Now imagine that the sections only contain precise age constraints at their tops and bottoms. In this case, even though the signals are trivially correlated, the location of the peak and trough of the sine in absolute time will be highly uncertain. With extremely ignorant priors on the age modeling, it's conceivable that the posterior distribution for the common proxy signal in absolute time may even have flat contours that completely encapsulate the amplitude of the sine. Ideally, the priors on the age modeling would ensure that the temporal structure of the posterior appropriately encapsulates the true structure. However, the figures that the authors present demonstrate that this is not the case for their model. For example, in Figure 8, the authors show that the posterior model nicely recovers the overall shape of true common proxy signal. However, the locations of peaks and troughs in absolute time seem (subjectively speaking) too tightly constrained, such that the posterior model significantly deviates from the true model at the locations of most major peaks and troughs. This result seems to be due entirely to the age modeling, for which the priors appear to be \*too\* informative. This subjectively described behavior can be quantified with a reliability diagram (see a subsequent comment).

The stratigraphic sections used for the inference in Figure 8b include a number of 'extra' depositional and correlative age constraints (illustrated in Figure 8a), in addition to the minimum and maximum ages shared by all sections. This additional information reduces uncertainty in the age of each sample, both directly (for the section hosting the age constraint) and through correlation (for all other sections). As a result, the posterior envelopes are narrower than if only minimum and maximum ages were available. The inferences in Figures 9 and 11 of the main text, which use sections with only minimum and maximum age constraints, provide more appropriate analogs to your imagined scenario. In both examples, the posterior envelopes are significantly wider than in Figure 8.

To facilitate direct comparison, Figure R1 juxtaposes the inference with 'extra' ages against a version that only includes minimum and maximum age constraints for each section (all other parameters are unchanged). As expected, the 95% envelope of the inference is significantly flatter when only minimum and maximum age constraints are considered (Figure R1b). The 95% envelope in Figure R1b is not completely flat because the depositional histories required to produce the most extreme solutions (i.e., where the entire signal is compressed to a small fraction of the available time) have very low but non-zero prior probabilities. Thus, these extreme solutions exist but often fall outside of the 95% contours.

[Figure]

**Figure R1.** Comparison between inferences that include 'extra' intermediate age constraints *(a)* versus only minimum and maximum age constraints *(b)*. Each additional age helps to constrain the timing of the proxy signal, reducing uncertainty in the posterior.

We disagree that the posterior model "'significantly deviates' from the true model at the locations of most major peaks and troughs". In this example, the true signal is consistently captured by the 95% envelope of the inference, meaning that the model has successfully recovered the correct answer. Given that the age models are appropriately uninformed (see discussion below), the posterior likelihood of the true signal sometimes may be low (i.e., not captured by the 33% or 66% envelopes); better constraining the signal in time ultimately requires more absolute age constraints. Moreover, there is no strict requirement that the true signal should always be fully encapsulated by the posterior. For example, if a given excursion peak is not observed in any section (due to incomplete preservation or low sampling resolution), then the posterior may not capture that feature even if the signal is well-constrained in time.

All this discussion brings me to the main point, which is that the authors need to more critically consider the prior age modeling. The authors state that they (line 138) "construct prior age models with the goal of imposing no limits on sedimentation rate between age constraints," and yet in Figure 3c they show the distribution resulting from their prior modeling approach.

The statement that we "impose no limits on sedimentation rate" was intended to convey that the prior encapsulates the full range of possible sedimentation rates (i.e., from nearly instantaneous deposition of the entire sediment package to continuous deposition over the entire time period), not that all possible sedimentation rates are equally likely. To clarify our intention, we added the following text (lines 141-143):

> We construct these prior age models with the goal of imposing no limits on sedimentation rate between age constraints, meaning that the possible depositional histories for each section range from highly episodic to uniform (Fig. 3c). **The prior likelihood of different depositional histories also should reflect our knowledge about**

**how sediment accumulates in nature: namely, that extremely large and rapid depositional events are rare compared to more gradual sedimentation (Sadler, 1981).**

What is this distribution? Sadler (who the authors cite) demonstrated that the concept of sedimentation rate is only relevant at a timescale of interest, since the Sadler Effect shows that sedimentation rates decrease as a power law with averaging timescale. At a particular timescale of interest (within an order of magnitude or so), Sadler also demonstrated that sedimentation rates follow a log normal distribution. Figure 3c does not appear to be log normal. The ad-hoc approach that the authors have taken with the shift and scale parameters was probably motivated by modeling convenience, but it muddies the waters in terms of incorporating empirical statistical information about sedimentation rates.

Sadler's finding that sedimentation rates follow a log-normal distribution is a useful statistical description of a natural phenomenon, not a first principle. Our age modeling is merely a different approach to emulating the same underlying phenomenon: episodic deposition and erosion of sediment. The scale-shift parametetrization, far from being ad-hoc, is designed to mimic this physical process. In brief, the 'scale' parameter reflects that a given section may have been deposited extremely rapidly (highly compressed solutions corresponding to small scale values), or may instead have been deposited gradually over a more extended period of time. The 'shift' parameter simply slides this 'depositional window' forward and backward in time. Sample ages then are drawn from a uniform distribution within this window, which encodes the intuition that the within-section relationship between stratigraphic height and time may be similarly irregular.

While we acknowledge it is difficult to build intuition for exactly how the scale and shift parameters ultimately shape the prior, the resulting prior age models demonstrably 1) accomplish our goal of encompassing all geologically possible depositional histories, and 2) possess the same essential characteristics as Sadler's model. The first point is illustrated by the wide contours of the prior age-height models (Figure 3c) and by the individual sample age priors, which span the entire interval between the bounding geochronological ages (Figure R2).

[Figure]

**Figure R2.** Prior age model for the example section in main text Figure 3c. Each shaded blue distribution represents an individual sample (i.e., $\delta^{13}$C observation). Each sample age prior spans the entire range between the minimum and maximum age constraints, with lower probability density assigned to geologically unlikely solutions that require extremely rapid aggradation. For example, the lowermost (oldest) sample is more likely to be similar in age to the basal (maximum) geochronological age constraint than to the overlying minimum age constraint.

It also is straightforward to establish that our prior age models are broadly consistent with Sadler's statistical model. Specifically, Figure R3 shows that the prior sedimentation rate distribution from Figure 3c is approximately log-normal (i.e., normally distributed on a log scale), albeit right-skewed. The left tail of the distribution is truncated because, given fixed 1 meter spacing between samples and a maximum elapsed time of 100 Myr, the minimum possible apparent accumulation rate (calculated between pairs of adjacent samples) is $\frac{1 \text{ m}}{100 \text{ Myr}} = 0.01$ m/Myr. In other words, our prior must diverge from Sadler's statistical model in order to exclude physically impossible values. The precise shape of the sedimentation rate prior will vary between sections, depending on both sample spacing and geochronological age constraints, but the distribution is always approximately log-normal and generally right-skewed. We also note that for any given pair of samples, the prior age model always yields a power law relationship between sedimentation rate and averaging timescale (i.e., forms a straight line on a log-log plot; Figure R4) – another key prediction of Sadler's statistical model. This power law relationship is the inevitable result of holding stratigraphic thickness constant while varying the time elapsed between samples. This statistical similarity suggests that our model is capturing the same fundamental features of the stratigraphic record as Sadler's; we have merely taken a different path to the same destination.

[Figure]

**Figure R3.** Log-normal fit to the prior sediment accumulation rate distribution from main text Figure 3c. Accumulation rates are calculated between adjacent samples. Note that our original Figure 3c used a $\log_e$ scale instead of the standard $\log_{10}$ scale; this error has been corrected.

[Figure]

**Figure R4.** Prior sediment accumulation rates for the synthetic example in Figure 3 of the main text. Sedimentation rates are calculated between all possible sample pairings. Because stratigraphic heights are fixed, the prior age models always plot along straight lines in log-log space (i.e., exhibit power law relationships).

In the manuscript, we added the following text to clarify our age modeling approach and establish its consistency with existing theory (lines 151-157):

**The resulting prior age models encompass all possible depositional histories, but assign lower prior probabilities to solutions that are geologically unlikely. For example, extremely rapid deposition of the entire section (the far right tail of the prior sedimentation rate distribution; Fig. 3c) is less likely than more gradual deposition. The prior age models also are consistent with Sadler's (1981) empirical model of how time is distributed in stratigraphy: sedimentation rates are approximately log-normally distributed (Fig. 3c), and each section's age model exhibits a power-law scaling between timespan and apparent sedimentation rate (i.e., the Sadler effect).**

I recommend that the authors reformulate their priors for the age modeling. Specifically, the authors should explicitly consider how the sedimentation rates they model probabilistically are tied to a timescale (as they must be), which itself might be a random variable in their model. If the authors think that proxy sample spacing may span timescales over multiple orders of magnitude, they should grapple with the implications that makes for prior sedimentation rate modeling throughout a sampled section. I recommend abandoning the ad-hoc approach, which imposes a poorly interpretable prior. Finally, the prior that is ultimately chosen should result in age models that yield *reliable* posterior models (see subsequent point).

Our model is intended to be maximally generalizable, such that it can be applied to data from a wide range of depositional environments that may have sedimentation rates spanning orders of magnitude, both within and between stratigraphic sections. Given this goal, it would be counterproductive to force the user to place a more restrictive prior on sedimentation rates within each section. This modeling choice is made clear in the text, and users that wish to adopt more informed or restrictive prior age models would be free to modify or expand the model in a direction useful for their specific application. Future versions of StratMC could allow users to provide custom prior age models – perhaps even linking our framework to independent Bayesian age modeling approaches such as Zhang et al. (2023), which incorporates information about subsidence trajectories in extensional basins – but this modification is beyond the scope of the 'base version' of the model introduced in this manuscript.

We also note that the prior age models are not a 'black box'; they can easily be inspected by the user. For example, functions in the `stratmc.plotting` submodule can be used to plot the prior sample age distributions, generate a Sadler plot, or view section age-height models. For a given data set, these tools help to build intuition for how our age modeling approach translates to more readily interpretable/traditional metrics such as sedimentation rate.

As an aside, I could imagine that this modeling framework might be able to introduce an intermediate hidden variable that captures the well-constrained, correlated component of the common proxy signal, which exists along a coordinate that has a monotonic relationship with both absolute age *and* stratigraphic height in each section. This internal representation separates the correlation from the age modeling problem. The task would then be to evaluate the likelihood of the monotonic maps between age and height for each section, mediated via this hidden variable. These mappings would be informed by the prior assumptions about sedimentation rate as well as potentially any stratigraphic information indicating disconformity, etc.

We appreciate the suggestion, but a central advantage of our modeling approach is that the correlation and age modeling problems are *not* separated. Our integrated approach is particularly essential for highly fragmentary data, where the common proxy signal may not have a 'well-constrained, correlated component' before the geochronology is considered.

Also note that it is already possible to incorporate stratigraphic information about e.g., disconformity/hiatus via creative definition of age constraints. For example, to model a hiatus with some minimum/maximum duration, one could simply define two uniform age constraints (representing the base and top of the disconformity surface) with upper and lower bounds that reflect any available geochronological age constraints and/or prior knowledge about the approximate age and duration of the hiatus. Section 2.4 touches on incorporating these types of 'undated' stratigraphic features (focusing on correlative surfaces), but our modeling framework is flexible enough to incorporate many other types of stratigraphic information.

**1.3 Quantifying model performance with reliability diagrams**

I appreciated the authors' mean signal likelihood metric, which captures the overall performance of their modeling approach in a single number. However, given that they are evaluating the performance of a probabilistic inference with respect to the truth, they should also utilize reliability diagrams, which show how the predicted distribution of values corresponds to the actually observed distribution. Bröcker and Smith (2007) (https://doi.org/10.1175/WAF993.1) provide a useful reference for constructing reliability diagrams with bootstrapped confidence intervals.

Thank you for the suggestion to investigate other tools for evaluating model performance. We experimented with reliability diagrams[**]; below, we also try a different tool that we think provides more insight. As a test, we constructed a reliability diagram for the inference in Figure R1b by plotting the observed frequency of $\delta^{13}$C values in the true signal (within $0.5‰$ bins) against their predicted frequency (combining the posterior predictions across all time steps) (Figure R5). Most observations fall near or slightly below the 1:1 line for a perfectly reliable model, as you predict below. Subjectively speaking, these deviations are minor, suggesting the model is satisfactorily reliable.

However, we do not think that a reliability diagram is necessarily an appropriate evaluation tool for our model. It is our understanding that reliability diagrams are designed to assess the performance of predictive models with discrete outcomes (e.g., predicting whether or not it will rain). By applying this concept to a continuous regression model, we obscure much of the nuance about how well the model is actually performing: for example, a prediction that deviates from the true value by only $1‰$ is better than a prediction that deviates by $5‰$, but the reliability diagram makes no such distinction. The reliability diagram also does not provide insight to when and where the model might be over- or underestimating uncertainty (e.g., near the edges of the signal, or at high vs. low $\delta^{13}$C values). Below, we consider a different assessment tool that is potentially more appropriate for continuous regression models.

[**] We were somewhat unsure of how to go about constructing a reliability diagram since we only really have one unique observation (the true signal). The best solution we could come up with was to combine the predictions and observations across all time slices, which allowed us to calculate an 'observed frequency', but yielded a reliabilty diagram that is poorly interpretable. Apologies if we misunderstood your suggestion, in which case further clarification is welcome.

[Figure]

[Figure]

**Figure R5.** *(a)* Reliability diagram for the proxy signal inference in Figure R1b, where each data point represents a different $\delta^{13}$C bin with a width of 0.5‰. Note that the outliers with high observed frequencies correspond to baseline $\delta^{13}$C values near 0‰. *(b)* Posterior and observed relative frequencies used to construct the reliability diagram in *a*.

I suspect the authors will find that the current formulation of their model underestimates the true signal at both the lower and upper prediction quantiles due to the afore-mentioned over-confidence in the absolute time location of the common proxy signal. That is, I expect that the reliability diagrams for the current modeling approach would have low slopes falling off of the 210 1:1 line for a perfectly reliable model, in which case hopefully a modified age modeling prior would improve reliability.

Altogether, your discussion points toward three more general concerns: 1) that we have not examined how well the $\delta^{13}$C observations are captured by the posterior, 2) that in some places, the true signal is not as likely as we would hope, especially at the peaks and troughs, and 3) that the posterior envelopes are not sufficiently wide. Previous discussion of our age modeling approach, combined with Figures R1 and R5, helps to resolve (3). To address the first two concerns, we constructed 'predicted 215 vs. observed' plots (Figure R6) for the inference in Figure R1b.

Figure R6a shows the relationship between the $\delta^{13}$C observations and the proxy signal inference, while Figure R6b examines the relationship between the true proxy signal and the proxy signal inference. In both cases, the data are concentrated around a 1:1 line, indicating the true and predicted values are usually approximately equal. However, performance declines slightly around both high and low $\delta^{13}$C values. Specifically, deviations from the 1:1 line around $-3$, $-1.5$, 3, and 5‰ correspond to 220 excursion peaks and troughs.

To examine why the model tends to underestimate $\delta^{13}$C around excursions, we designed a simple synthetic experiment (Figure R7). First, we generated synthetic $\delta^{13}$C observations using a sinusoidal proxy signal, and assigned small age uncertainties ($\pm 3$ Ma, $2\sigma$) to observations younger than 100 Ma and large age uncertainties ($\pm 50$ Ma, $2\sigma$) to older observations. Then, we performed Gaussian process (GP) regression through these data points using an exponential quadratic covariance function (as 225 in our stratigraphic model). The resulting proxy signal inference is both vertically and temporally offset from the true signal

[Figure]

**Figure R6.** Comparison between posterior proxy signal $\delta^{13}$C values and either the sample $\delta^{13}$C observations *(a)* or the true value of the $\delta^{13}$C signal *(b)*. Each x-axis bin (column) is normalized between 0 and 1. Recall that the $\delta^{13}$C observations were convolved with white noise to simulate natural variability, which contributes to the scatter in panel *a*.

around observations with large age uncertainties. While the true signal always is captured by the 95% envelope, it frequently falls outside of the interior envelopes. There are three mechanisms contributing to this behavior:

1. Large sample age uncertainties translate directly to large uncertainty in the timing of the signal, even if the structure of the signal (i.e., the relative timing and magnitude of excursions) is well-constrained. This means that if we are reconstructing a $+5‰$ excursion, the model will always infer that there is a peak at 5‰ *somewhere*, but the posterior distribution for the age of the peak will be wide, such that it may only occur at the correct time in 1% of solutions. Consequently, the posterior likelihood of the peak proxy value at any single time is low (because in most solutions, the peak is located elsewhere), causing the peak to appear 'flattened' or underestimated. In this example, the large uncertainties also 'drag' the signal to the left due to interactions between the sample age priors. Note that this behavior is significantly amplified because there is no maximum age constraint (as in our stratigraphic model), and due to the asymmetric uncertainties for old and young samples.

2. All stationary Gaussian process covariance kernels are mean-reverting. Put simply, the inferred proxy signal will trend back towards the mean during time periods with no observations. It is possible that when age uncertainties are very large, the model could 'cluster' the observations tightly in time, leaving observational 'gaps' elsewhere. Within these observational gaps, the signal will begin to revert to the mean on a timescale that depends on the GP lengthscale. In theory, mean reversion could contribute to peak underestimation, especially for particularly small or sparse data sets. Intervals of the posterior affected by mean reversion should also have particularly wide posterior envelopes, reflecting greater uncertainty in regions that are not well-constrained by data. In Figure R7, this 'ballooning' of uncertainty with

increasing distance from the data also explains why the upper contour of the 95% is higher than the maximum observed
$\delta^{13}$C value. For the experiments in our manuscript, interrogation of the posterior suggests that posterior data density is
generally high enough that mean reversion is not a significant factor. This is especially true for the inference in Figure
8, where relatively dense geochronological age consraints preclude 'highly compressed' solutions that would minimize
the temporal coverage of the data.

[Figure]

**Figure R7.** Gaussian process regression through $\delta^{13}$C observations with large age uncertainties ($\pm 50$ Ma, $2\sigma$) on the left and small age
uncertainties ($\pm 3$ Ma, $2\sigma$) on the right. Sample ages are modeled with Gaussian uncertainties. This toy example illustrates how large age
uncertainties can shift both the timing and $\delta^{13}$C of the posterior inference relative to the true signal.

In summary, closer interrogation of our results suggests that the model is performing as expected: the posterior consistently
captures the true signal, and the confidence in that inference decreases as information is removed (e.g., as age uncertainties are
increased). Model accuracy is demonstrated by the approximately 1:1 relationship between the observed and predicted values
(Figure R6). Deviations from this 1:1 line (e.g., around high and low $\delta^{13}$C values) are fundamentally related to large sample
age uncertainties, and further improving model performance ultimately requires reducing these uncertainties by incorporating
additional data.

Note: While a useful exercise, we did not add 'predicted vs. observed' plots to the figures in the manuscript since similar
conclusions about model performance can be drawn through visual inspection of the proxy signal inference.

In figure 9c, reliability diagrams may also reveal another slightly troubling result. The authors nicely show how the incorpo-
ration of multiple proxies significantly increases the synchronicity of the posterior with each true common proxy. However, the

confidence intervals, especially at the low and high tails, do not seem to sufficiently collapse to reflect the improved modeling.
260 Why might the model be overestimating uncertainty in the tails for inferences with more proxy systems?

Incorporating new information reduces uncertainty in the correlation among sections, which narrows the posterior age models and improves recovery of the true proxy signal. However, this reduction in uncertainty is ultimately limited by the density of geochronological age constraints. In other words, the improved correlation concentrates the posterior density around the true solution, but the posterior still has long tails due to uncertainty in the absolute age models.

**1.3.1 Comparison with Lee et al. (2023)**

While the authors do mention Lee at al. (2023) in the introduction, I think more can be done to compare the two modeling approaches. To my knowledge, Lee et al. (2023) provide the sole other Bayesian approach to simultaneous age modeling and (single) proxy correlation. Given that the authors are presenting exactly the same sort of model, they should be more explicit in acknowledging the similarities between the models and then highlighting the contributions they have made to this type of
270 modeling, namely:

1) prior age modeling that is not strictly tied to assumptions about deep sea sedimentation rates (although as previously mentioned, this approach needs to be better grounded in theory) and

2) multi proxy correlation (which is mentioned in section 4.2, but with insufficient context)

Section 4.2 needs to reference Lee et al. (2023), and there could be a couple more sentences highlighting the similarities
275 between the models either in the introduction or methodology. For instance, Lee et al. (2023) also permit inference of section-by-section offsets and variance scaling with respect to the inferred common proxy signal (albeit for a single proxy).

We have added the following comparison to Lee et al. (2023) in Section 4.2 (lines 542-551):

**Currently, the only comparable Bayesian chemostratigraphy algorithm is the BIGMACS model developed by Lee et al. (2023). BIGMACS is similar to StratMC in that it both correlates geochemical proxy profiles**
280 **and constructs section age models. However, unlike StratMC, its age modeling approach relies on prior information about sedimentation rates in deep-sea cores that generally is not available for more ancient and incomplete shallow-water stratigraphies. BIGMACS also does not allow for simultaneous correlation of multiple geochemical proxies (e.g., $\delta^{18}$O and $\delta^{13}$C). BIGMACS does include per-section proxy offsets, and it uses Gaussian process regression to construct a composite proxy 'stack'. Importantly, the stack construc-**
285 **tion and correlation/age modeling steps are performed iteratively rather than simultaneously, meaning that sections are aligned to the current stack – essentially the mean and variance of the data over time, excluding outliers – rather than to a common 'hidden' proxy signal. BIGMACS also requires the user to provide an initial alignment target, necessitating some prior knowledge of the signal to be reconstructed.**

I think the authors should consider applying their model to the same d18O and radiocarbon dataset modeled by Lee et al.
290 (2023) (perhaps just the Deep North Atlantic dataset, for example) as an application with real-world data and an opportunity

for direct inter-model comparison. Several if not all of the cores utilized by Lee et al. (2023) in that stack have other proxy measurements (d13C, elemental concentrations, etc.) that could be utilized by the authors' new approach.

Thank you for the suggestion, but more detailed inter-model comparison is beyond the scope of the current manuscript. First, applying our model to the relatively large Lee et al. (2023) data set would consume weeks of computation time, which is not currently feasible. Second, applying our model to relatively well-behaved recent deep-sea records may be misleading to readers. As you noted above, the Lee et al. (2023) model is specifically designed for such deep-sea environments, where we have more prior knowledge about sediment accumulation rates, and thus considering those sedimentation rates explicitly in the age model is the optimal choice. Our model will inevitably produce a proxy signal inference with wider uncertainty envelopes (due to the less restrictive prior age models, which make minimal assumptions about sedimentation rates) than Lee et al. (2023), which may misleadingly imply poor model performance when in fact the issue is poor model choice.

**1.3.2 "Non-uniform" depositional histories**

The treatment of "episodic" sedimentation could also be better grounded in the theory of time's distribution within stratigraphy. The Sadler Effect arises due to the power law distribution of hiatus within stratigraphy; the distribution of hiatus therefore dictates apparent sedimentation rates. Hiatuses result from the dynamics (autogenics) of sedimentation as well as processes such as sea level, tectonics, etc. A critical timescale is the compensation timescale: below this timescale, stratigraphy is incomplete, which approximately corresponds to the "episodic" realm described by the authors. Beyond this timescale, stratigraphy is complete (the "continuous" regime), up until hiatuses resulting from longer timescale processes such as tectonic modifications of basin accommodation. What the authors refer to as "temporal noise" and "episodic sedimentation" are in fact the statistical structure of hiatus in stratigraphy, which results in stratigraphic incompleteness at short and long timescales.

We appreciate this useful background, but would like to point out that episodic sedimentation and erosion are the basic physical phenomena that give rise to the statistical models you have described, not the other way around. The review you cite below, for example, states that "The overall low completeness of stratigraphy results from spatial and temporal intermittency of transport" (Paola et al., 2018). We attempted to avoid using stratigraphy jargon to ensure the manuscript is accessible to a more general audience, but our terminology is not incorrect.

This background brings me to the main point of this comment, which is that the authors need to take care that they are realistically modeling stratigraphy as best as we understand it when constructing their synthetic examples. For instance, by modeling the elapsed time between approximately evenly-spaced (in space) samples as a gamma distribution, do the resulting height increments ('the devil's staircase' shown in the right panel of Figure 6b) have a truncated/exponentially tempered power law distribution for small values of $k$, as we expect (Ganti et al. 2011)? How does the truncation of the power law (i.e., the compensation timescale) depend on the value of $k$? The authors should establish the theoretical connections between their current stratigraphic synthesis protocol and the relevant quantities in our current understanding of time's distribution in stratigraphy (such as the compensation timescale, the power law distribution of hiatus in stratigraphy, which truncates at intermediate (post

compensational) timescales). Paola et al. (2019) provide a great review of these concepts (https://doi.org/10.1146/annurev-earth-082517-010129).

325     Alternatively, the authors could reformulate how they generate their synthetic stratigraphies. For example, rather than sampling time increments according to a gamma distribution, which appears to be an arbitrary decision unmotivated by theory, they could instead simulate stratigraphies with varying compensation timescales and stratigraphic completeness (i.e., hiatus power law exponents), which are then sampled regularly (or not) in space. The results of Section 3.3.2 would then be much easier to interpret with respect to the theoretical framework that exists for stratigraphy; perhaps the authors could modify Figure 13 to

330     reflect various values of the hiatus power law exponent (completeness) and compensation timescale.

    Thank you for pointing out that we failed to reference appropriate theory in this section. The choice of a gamma distribution was not arbitrary, but rather a simplification of the compound Poisson-Gamma age-depth model of Haslett and Parnell (2008), which is the basis for the popular Bayesian age-depth modeling package Bchron. The compound Poisson-Gamma model treats the age gaps as gamma distributions and the position gaps as exponential. Here, we only borrow the gamma component

335     to simulate age gaps, omitting the position component because our model only considers the stratigraphic order of samples (superposition), not their heights. Thus, the sample heights are, for the purposes of testing our inversion approach, essentially arbitrary.

    In the text (lines 270-271), we added language to highlight this connection:

    Depositional histories ranging from continuous to episodic are simulated following the procedure in Fig. 6. First,

340     we define a synthetic $\delta^{13}$C signal from 130 to 100 Ma (Fig. 6a). Then, we translate this signal to four stratigraphic sections (each with 30 samples) by modeling the time elapsed between samples, $\Delta t$, and the stratigraphic height between samples, $\Delta h$, as gamma distributions (Fig. 6b). **This parameterization is a modification of the compound Poisson-Gamma chronology model (Haslett and Parnell, 2008).**

    The gamma distribution has a thinner (exponential) tail than a power law (e.g., Pareto) distribution, making extremely large

345     age jumps comparatively unlikely. While we acknowledge the power law distribution of hiatus in stratigraphy (e.g., Schumer et al., 2011), the choice between a thin or heavy-tailed jump distribution does not change the take-home message of this section: that the model can take data from many incomplete sections and recover a signal that is similar to what might be preserved in one complete section.

    We concur that framing our experiments in terms of concepts like the compensation timescale and hiatus power law exponent

350     may improve interpretability for readers who are already well-versed in stratigraphy theory, but such exposition would likely muddy the waters for our more general target audience. Properly defining these concepts for readers would also require adding substantial – and ultimately tangential – theoretical background to an already lengthy manuscript, detracting from the main point: describing and validating our chemostratigraphy model.

    Since the concept of stratigraphic completeness is both simple and important, we have adjusted our language to highlight

355     the relationship between the episodicity of deposition and completeness. This link remains qualitative; because completeness

must be quantified with respect to a particular timescale, making the specific values somewhat unimportant for our purposes, we opt to retain our 'low-k to high-k' scale in Figure 13.

Finally, while we think highlighting this conceptual link is valuable, drawing generalizable conclusions about the relationship between proxy signal recovery and completeness would require a more involved experimental design (multiple trials with simulations spanning various compensation timescales, completeness levels, etc.), which is beyond the scope of this methods paper. This is an intriguing area for future research that likely would constitute a stand-alone manuscript.

Revisions to Section 3.1.1 (line 273):

The shape parameter, $k$, of the gamma distribution controls whether the samples are unevenly (low $k$) or uniformly (high $k$) spaced. Different depositional histories are modeled by varying $k$ for the $\Delta t$ distribution between $0.1$ and $10$; **stratigraphic completeness (i.e., the proportion of time preserved in the strata; Sadler, 1981) increases with $k$.**

Revisions to Section 3.3.2 (lines 393-409):

Deep-water environments with relatively constant sedimentation rates classically are considered to be the most reliable archives of past proxy change. However, since a large fraction of deep-sea sediments older than $\sim$200 Ma have been subducted at continental margins, reconstructions of marine proxy signals prior to the mid-Mesozoic instead rely primarily on sediments deposited in shallow-water environments, where **more episodic deposition leads to low stratigraphic completeness**. Here, we use our model to consider how these shallow-water recon-structions may stack up to those based on more **complete** deep-sea records. We build stratigraphic sections with depositional histories ranging from episodic ($k = 0.1$) to uniform ($k = 10$) following the procedure detailed in Sect. 3.1.1. For each experiment, we apply our inference model to between four and ten sections generated using the same $k$ value.

Our model validates the intuition that continuous deep-sea records are preferable to **less complete** shallow-water records. For a fixed number of stratigraphic sections, signal recovery generally improves as $k$ increases (i.e., as sedimentation becomes less episodic, **increasing completeness)** (Fig. 13c). For each $k$ value, signal recovery also improves as additional sections are considered. Still, all hope is not lost for stratigraphers working in ancient shallow-water strata: signal recovery for models that include a large number of **highly incomplete (low-$k$)** sections is comparable to signal recovery for models with a lower number of **complete (high-$k$)** sections (Fig. 13b). In other words, quantity can compensate for quality. For example, the mean signal likelihood for the 2-section model with $k = 10$ is $0.20$, compared to $0.14$ for the 2-section model with $k = 0.5$ and $0.21$ for the 10-section model with $k = 0.5$. As deposition becomes increasingly episodic (lower $k$), a greater number of sections is required to achieve comparable performance.

**1.4 Technical Comments**

– In Equation 4, the notation seems imprecise. Is it not in fact the evaluation of the posterior over $\theta$ at the true proxy value? $P_{\theta_{f(t_n)}}(g(t_n))$.

Thank you for pointing this out; our notation has been updated.

– Fig 1

   – would be nice to annotate the relevant parts with the notation introduced in Equation 1

   We prefer to minimize use of mathematical notation for clarity – Figure 1 is meant to provide a broad overview of the model structure that can be understood without closely reading the rest of the methods section. The components of Figure 1 also do not directly map to the terms in Equation 1. For example, while the model 'outputs' are equivalent to the posterior, $P(\theta|\mathcal{D})$, the model inputs are distinct from the model prior, $P(\theta)$, and the model components are related to both the prior and the likelihood.

   – empty box under model seems to serve no purpose

   The empty box simply represents the full model, which includes the sub-components connected by dashed lines. We find this configuration more aesthetically consistent with the model input and output panels than e.g., removing the box and keeping only the 'Model' label.

– fig 5

   – would be easier to compare panels a and b if the axes in panel a were flipped

   Thank you for the suggestion; however, we prefer to put age on the x-axis in our subsequent results figures, and have kept Figure 5 as-is for internal consistency.

– line 363: might be worth clarifying that white noise is independent, identically distributed, zero mean

The subsequent sentence states that: "White noise is generated using a normal distribution with $\mu = 0$ and $\sigma$ between $0.5$ (the low-noise endmember) and $5.0$ (the high-noise endmember)". We have also added the term 'Gaussian noise' to the previous sentence for extra clarification (line 279). By definition, white Gaussian noise is independent and identically distributed.

– line 484-485: This sentence minimizes the importance of the age modeling procedure for the construction of the common proxy signal...it's not really like the age modeling is a byproduct of the authors' model. It's an integral part of the inference.

Our intention was not to minimize the role of the age modeling in the inference, but rather to emphasize that signal reconstruction is the model's main purpose, from the end-user's point of view. In contrast, most existing models aim only to correlate sections or construct age models. We have adjusted our language to clarify our intended meaning (line 500):

Correlation and age model construction are **integral parts** of the signal reconstruction process, but are not the main objective.

420    – line 567, 596: siliclastic -> siliciclastic

Thank you; typos have been corrected.

**2   Response to RC2**

The authors have developed a new Bayesian statistical framework capable of inferring common (background) proxy signals that result in numerous local expressions measured from stratigraphic sections. This is a huge need for the Earth history community and a major step forward. Although I have a few questions detailed below, I found this contribution compelling and well-presented, and I recommend that this work be published. The authors should feel free to adopt or ignore my suggestions at their discretion. Overall, I greatly appreciate the quantitative approach that the authors have developed and look forward to seeing it published soon.

Thank you for the thoughtful and encouraging comments on our work. We address your questions below, and think that implementing your suggestions has improved the manuscript.

Line 43: I also recommend citing Trower et al. (2024; Geophysical Research Letters) here because they present an additional dataset and work to constrain the diurnal engine effect like Geyman and Maloof (2019; PNAS).

We have added the Trower et al. (2024) citation (line 43).

Lines 47-51: Hagen (2024, Geochemistry, Geophysics, Geosystems) discusses many of these quantitative approaches and could be good to cite here.

Thank you for highlighting this interesting review; we now cite Hagen (2024) in this section (line 49).

Line 55: Dynamic time warping has been used to align more than just isotopic profiles in stratigraphy, including borehole well data (Baville et al., 2022; Marine and Petroleum Geology; Sylvester, 2023; Basin Research), paleomagnetic data (Peti et al., 2020; Geochronology; Reilly et al., 2023; JGR Solid Earth; Hagan et al., 2020; Geophysical Journal International), ice core data (Hagan and Harper, 2023; Annals of Glaciology), fish otolith geochemical profiles (Lilkendey et al., 2025; Global Change Biology), and more.

Thank you for pointing out this previous work. We have expanded our literature review accordingly, both here (lines 53-57) and in Section 4.2 (Comparison with alternative approaches; lines 510-512).

Lines 133–137: Of course, this assumption is necessary for this kind of approach, and but I wonder how useful this approach would be if the records in question are all significantly affected by local or diagenetic processes? Is it possible to still arrive at a background signal if all records are highly perturbed? Could you design a synthetic experiment of some kind where input records are progressively perturbed, either with random noise or a localized trend, and see if there is a threshold where this

approach can no longer arrive back at the 'true background' signal? I see that you start to get at this in Section 3.1 below, but I think a specific test for this 'amount' of local variability or diagenesis, etc. needed to cause a failure would be valuable.

450 This is a valuable but complicated line of inquiry. Since diagenesis and local variability can alter proxy values in a number of different ways – for example, different modes of diagenesis might increase variance, decrease variance, and/or impart stratigraphic trends in proxy values – it is difficult to place a general threshold on 'how much' noise is required to entirely obscure the proxy signal. However, we can use our experiments in Section 3.3.1, where we add zero-centered white noise with amplitudes between 0.5 and 5.0‰ to the proxy observations, to consider scenarios where diagenesis or local variability

455 manifests as random noise. Figure 12d shows that signal recovery deteriorates steadily until the variance of white noise reaches ~2‰ and then stabilizes around a 'mean signal likelihood' value of 0.175. In general, oscillations with magnitudes lower than the amplitude of added white noise cannot be recovered by the model. For example, the negative 2.5‰ excursion around 120 Ma is largely lost when $\sigma_{\text{noise}} = 1.5$‰ (Figure 12b). Thus, we can broadly say that the proxy signal becomes 'unrecoverable', in the sense that the details of the structure are largely lost (as illustrated in Figure 12b), when the $2\sigma$ amplitude of the added

460 noise exceeds that of the common signal.

Although this finding is not fully generalizable – for example, the model's ability to 'see through' high-amplitude noise could be bolstered by additional geochronological constraints – we think it provides a useful baseline. As such, we have added the following sentence to the manuscript (lines 382-384):

As the amplitude of added white noise increases, the model is only able to resolve higher-amplitude and longer-

465 term (lower-frequency) features of the proxy signal. The confidence envelopes of the signal inference become wider and smoother with increasing $\sigma_{\text{noise}}$, effectively 'blurring' the signal as progressively larger isotopic excursions are obscured (Fig. 12b). In this example, signal recovery initially declines as $\sigma_{\text{noise}}$ increases and stabilizes above $\sigma_{\text{noise}} = 2$ (Fig. 12d). **These experiments suggest that signal recovery deteriorates dramatically when the amplitude ($2\sigma$) of random noise meets or exceeds the amplitude of the common proxy signal.** In gen-

470 eral, however, the model's resolving power is sensitive to both the amplitude of added noise and the density of geochronological age constraints, where more age constraints may help to combat the blurring effect of proxy noise.

Line 304: The multi-proxy approach is so important and it's great to see someone trying to tackle this quantitatively!
Thank you! We look forward to tackling real multi-proxy geologic data with this approach.

475 Line 500: What if the data being correlated do not have any absolute age constraints? Is the main advantage then not having to designate a 'backbone' section?
Yes; our approach retains this advantage regardless of whether any absolute age constraints are available. In the absence of absolute age constraints, the minimum and maximum ages for all sections could simply be set to arbitrary values (e.g., 0 and 100) with large uncertainties. An additional advantage is that undated correlative constraints (e.g., sequence or biozone

480    boundaries) could still be incorporated by e.g., modeling each correlative horizon as a uniform distribution bounded by the same arbitrary minimum and maximum values.

Line 723: Would it be possible to use this approach with stratigraphic time-series data that are not geochemical in nature? For example, could you use gamma-ray logs or relative grain size, etc.? It would be interesting to be able to use both physical and geochemical parameters for correlation, at least in more local, intrabasinal scenarios.

485    Absolutely, thank you for pointing out this additional use case. We have added the following sentence to Section 4.2 (lines 532-533):

**While our experiments focus on correlation of geochemical proxies, our model can be applied to any quantitative stratigraphic data (e.g., relative grain size measurements, paleomagnetic data, gamma ray logs).**

Again, I believe that this is an important contribution, that it should be published, and I enjoyed reading it. Thank you for the opportunity to interact with your work.

490    Thank you again for taking the time to review our work; we appreciate your suggestions and encouraging feedback.

**References**

Hagen, C. J.: Quantitative and Nuanced Approaches Elucidate Carbon Isotope Records, Geochemistry, Geophysics, Geosystems, 25, e2024GC011718, https://doi.org/10.1029/2024GC011718, e2024GC011718 2024GC011718, 2024.

Haslett, J. and Parnell, A.: A simple monotone process with application to radiocarbon-dated depth chronologies, Journal of the Royal Statistical Society: Series C (Applied Statistics), 57, 399–418, https://doi.org/10.1111/j.1467-9876.2008.00623.x, 2008.

Lee, T., Rand, D., Lisiecki, L. E., Gebbie, G., and Lawrence, C.: Bayesian age models and stacks: combining age inferences from radiocarbon and benthic $\delta^{18}$O stratigraphic alignment, Climate of the Past, 19, 1993–2012, https://doi.org/10.5194/cp-19-1993-2023, 2023.

Paola, C., Ganti, V., Mohrig, D., Runkel, A. C., and Straub, K. M.: Time Not Our Time: Physical Controls on the Preservation and Measurement of Geologic Time, Annual Review of Earth and Planetary Sciences, 46, 409–438, https://doi.org/10.1146/annurev-earth-082517-010129, 2018.

Sadler, P. M.: Sediment accumulation rates and the completeness of stratigraphic sections, The Journal of Geology, 89, 569–584, https://doi.org/10.1086/628623, 1981.

Schumer, R., Jerolmack, D., and McElroy, B.: The stratigraphic filter and bias in measurement of geologic rates, Geophysical Research Letters, 38, https://doi.org/10.1029/2011GL047118, 2011.

Trower, E. J., Hibner, B. M., Lincoln, T. A., Dodd, J. E., Hagen, C. J., Cantine, M. D., and Gomes, M. L.: Revisiting Elevated $\delta^{13}$C Values of Sediment on Modern Carbonate Platforms, Geophysical Research Letters, 51, e2023GL107703, https://doi.org/10.1029/2023GL107703, 2024.

Zhang, T., Keller, C. B., Hoggard, M. J., Rooney, A. D., Halverson, G. P., Bergmann, K. D., Crowley, J. L., and Strauss, J. V.: A Bayesian framework for subsidence modeling in sedimentary basins: A case study of the Tonian Akademikerbreen Group of Svalbard, Norway, Earth and Planetary Science Letters, 620, 118317, https://doi.org/https://doi.org/10.1016/j.epsl.2023.118317, 2023.